



# Mapping snow depth at very high spatial resolution with RPAS photogrammetry

Todd A. N. Redpath[1,2], Pascal Sirguey[2], Nicolas J. Cullen[1]

[1]Department of Geography, University of Otago, Dunedin, 9016, New Zealand
[2]National School of Surveying, University of Otago, Dunedin, 9016, New Zealand

*Correspondence to*: Todd A. N. Redpath (todd.redpath@otago.ac.nz)

**Abstract.** Dynamic in time and space, seasonal snow represents a difficult target for ongoing *in situ* measurement and characterisation. Improved understanding and modelling of the seasonal snowpack requires mapping snow depth at fine spatial resolution. The potential of remotely piloted aircraft system (RPAS) photogrammetry to resolve spatial variability of snow

depth is evaluated within an alpine catchment of the Pisa Range, New Zealand. Digital surface models (DSM) at 0.15 m spatial resolution in autumn (snow-free reference) winter (02/08/2016) and spring (10/09/2016) allowed mapping of snow depth via DSM differencing. The consistency and accuracy of the RPAS-derived surface was assessed by the propagation of check point residuals from the aero-triangulation of constituent DSMs, and via comparison of snow-free regions of the spring and autumn DSMs. The accuracy of RPAS-derived snow depth was validated with *in situ* snow probe measurements. Results for snow free

areas between DSMs acquired in autumn and spring demonstrate repeatability, yet also reveal that elevation errors follow a distribution substantially departing from a normal distribution, symptomatic of the influence of DSM co-registration and terrain characteristics on vertical uncertainty. Error propagation saw snow depth mapped with an accuracy of ±0.08 m (90% c.l.). This is lower than the characterization of uncertainties on snow-free areas (±0.13 m). Comparisons between RPAS and *in situ* snow depth measurements confirm this level of performance of RPAS photogrammetry, while also highlighting the influence of

vegetation on snow depth uncertainty and bias. Semi-variogram analysis revealed that the RPAS outperformed systematic *in situ* measurements in resolving fine scale spatial variability. Despite limitations accompanying RPAS photogrammetry, which are relevant to similar applications of surface and volume change analysis, this study demonstrates a repeatable means of accurately mapping snow depth for an entire, yet relatively small, hydrological basin (~0.4 km$^2$), at very high resolution. Resolving snowpack features associated with re-distribution and preferential accumulation and ablation, snow depth maps

provide geostatistically robust insights into seasonal snow processes, with unprecedented detail. Such data will enhance understanding of physical processes controlling spatial distribution of seasonal snow, and their relative importance at varying spatial and temporal scales.





## 1 Introduction

Water storage within a snowpack is a function of snow depth and density. Seasonal snow provides a globally important water resource (Mankin et al., 2015; Sturm et al., 2017), which is highly variable in space and time (Clark et al., 2011). Because snow depth exhibits higher spatial variability than snow density, advances in measuring snow depth at high spatial resolution offers promise for improved estimates of snow water equivalent (SWE) (Harder et al., 2016).

Difficulties associated with collecting field observations hamper in characterising and understanding spatial variability in snow depth, and in turn improving spatially distributed modelling of seasonal snow. While insight can be gained via modelling at moderate to large scales (Winstral et al., 2013), resolving the fine-scale variability and its controlling processes remains limited by the ability to capture such variability in the field (Clark et al., 2011). The labour intensive nature of collecting widespread *in situ* measurements of snow depth has resulted in a focus on quantifying snow distribution at, or near, to the time of peak accumulation (Kerr et al., 2013), which may limit inferences regarding controlling processes operating earlier in the accumulation season.

With biweekly temporal resolution, Anderson et al. (2014) gained substantial insights into physical controls on seasonal snow processes, albeit with a dependence on statistical scaling to relate transect scale observations to basin scale processes. Alternatively, the nature of automated snow measurement instrumentation often precludes continuous *in situ* measurement across networks sufficiently dense to characterise fine scale spatial variability. Kinar and Pomeroy (2015) provide a comprehensive review of instrumentation and techniques for measuring snow depth and characterising snow packs. In summary, while instrumentation and methodologies exist for obtaining accurate, and temporally continuous, measurements of snow depth and related snowpack properties at point locations, adequately resolving the high spatial variability of snow depth remains a challenge. This is exacerbated by local field conditions, such as exposure to wind or the complexity of the topography and vegetation increasing further the spatial variability in snow depth (Clark et al., 2011; Kerr et al., 2013; Winstral and Marks, 2014).

Remote sensing has provided substantial advances in quantification of seasonal snow variability, with imaging sensors supporting spatial and temporal resolutions that allow a range of scales to be explored. Space-borne satellite imagers, particularly optical sensors, provide a synoptic view and accompanying step-change capability in capturing properties of snow covered areas, although trade-offs exist between competing resolutions (Dozier, 1989; Nolin and Dozier, 1993; Hall et al., 2002, 2015; Sirguey et al., 2009; Rittger et al., 2013). For example, the MODerate resolution Imaging Spectroradiometer (MODIS) permits near-daily mapping of snow covered area (SCA) at continental to global spatial scales, although relatively coarse spatial resolution limits the inferences that can be made regarding fine scale spatial variability and leads to uncertainty in subsequent applications, such as assimilation into hydrological models. The advance of geostationary meteorological satellites such as Himawari 8 & 9 sees comparable spatial resolution to MODIS acquired in near real-time (Bessho et al., 2016). Contemporaneously, multispectral sensors such as Sentinel-2 and Landsat 8 continue to improve the temporal resolution of imagery suitable for mapping snow at resolutions of 10 – 30 m (Malenovský et al., 2012; Roy et al., 2014). Passive and





active microwave sensors offer the capacity to retrieve estimates of snow water equivalent directly from space-borne platforms, but also suffer substantial limitations, including coarse spatial resolution in the case of passive microwave sensors, and complexities in successfully processing snow signals and accounting for complex terrain in the case of both passive and active sensors (Lemmetyinen et al., 2018).

Despite the progress in mapping SCA, reliable determination of snow depth, particularly in complex terrain, remains challenging. Modern, very high resolution stereo-capable imagers show promise for retrieving snow depth over large areas, from space, although the influence of topography on uncertainties, and complications introduced by shadows in alpine terrain demand attention (Marti et al., 2016).

Advances in Light Detection and Ranging (LiDAR) technologies have become increasingly relevant for measurement of
snow depth, firstly from air (Deems et al., 2013) and more recently from space-borne platforms (Treichler and Kääb, 2017). Of the three modes of LiDAR data capture, Terrestrial Laser Scanning (TLS) (e.g., Revuelto et al., 2016) offers the best performance in terms of precision and accuracy. TLS can resolve snow depth at a fine scale across relatively large areas, but remains hampered by view-obstruction in complex terrain, and logistical challenges of placing equipment *in situ* may limit deployment. Airborne LiDAR provides a balance of spatial resolution and accurate surface elevation measurement, but high
financial costs and logistical challenges impair regular capture. Treichler and Kääb (2017) assessed ICESat LiDAR data, which is designed primarily for measuring surface elevations over polar regions, for measuring seasonal snow depth in sub-polar southern Norway. Despite reasonable estimates of snow depth, measurements were accompanied by relatively large errors for most temperate locations. ICESat measurements are further limited by their punctual nature and footprint, yielding a relatively sparse and coarse spatial distribution, in turn complicating inferences about spatial variability.

Merit remains in characterising fine scale variability in snow depth distribution beyond the typical spatial scales of *in situ* measurements. Overcoming this challenge will enhance efforts to understand the processes controlling spatial variability of snow distribution at the basin scale, and improve understanding of the relative importance of meteorological and topographic controls of seasonal snow distribution.

Recent technological advances, including the miniaturisation of imaging and positioning sensors, as well as improvements
in battery power and autonomous navigation have significantly lowered the barriers associated with remotely piloted aircraft system (RPAS) operation (Watts et al., 2012). This, combined with ever-increasing computing power and significant improvements in machine-vision for dense photogrammetric reconstructions (Hirschmuller, 2008; Lindeberg, 2015) provide new opportunities to map small areas photogrammetrically at very high resolution in a temporally flexible, on-demand, fashion using RPAS. Examples of RPAS use related to mapping snow depth are promising, but tend to apply to sub-basin scales and
to not fully characterise the uncertainty associated with measurements (Vander Jagt et al., 2015; Bühler et al., 2016; De Michele et al., 2016; Harder et al., 2016).

Determination of snow depth via RPAS photogrammetry relies on the principal of differencing between temporally subsequent surfaces, provided by point clouds or digital surface models (DSM) (Vander Jagt et al., 2015; Harder et al., 2016). A snow-free surface provides a reference dataset for absolute snow depth, while changes in snow distribution through winter



can be assessed by comparing surfaces obtained while snow cover is present in the basin. Because changes in snow depth through time, either through processes of accumulation, ablation, or re-distribution may be subtle, the repeatability and vertical accuracy achieved by photogrammetric modelling is paramount.

The aim of this paper is to test a methodology for retrieving snow depth via RPAS photogrammetry and evaluate the performance, limitations and usefulness of this approach. Associated challenges include minimising spatial uncertainties sufficiently to reliably detect changes in snow depth over time, with a decimetre level of vertical accuracy targeted, while also reducing the need and complication of extensive *in situ* collection of ground control points (GCPs). Achieving this will resolve spatial variability in snow depth with improved detail compared to traditional methods, supporting improved insights into snowpack evolution. To this end, a campaign of winter RPAS-based photogrammetric surveys of a small alpine basin in the Pisa Range, New Zealand, was undertaken.

The paper describes the field site, field and photogrammetric methods, as well as the approach to quality and accuracy assessment before presenting the results both in terms of the photogrammetric derivation of snow depth and the assessment of the quality and accuracy of derived snow depth maps. The discussion addresses the performance of RPAS photogrammetry in this context, sources and nature of the associated uncertainty as well as pitfalls and limitations that were encountered, before demonstrating the insight that RPAS-derived data can provide for the study of seasonal snow. While primarily exploring and assessing the potential of RPAS photogrammetry for measuring seasonal snowpack, this study has broader implications for the wider field of modern close-range photogrammetry, as typically implemented from low cost, unmanned systems. Through scrutinising the performance of the RPAS system and associated software, and investigating potential sources of uncertainty, beyond those inherent in the aero-triangulation itself, this paper demonstrates some of the limitations and pitfalls that may affect RPAS photogrammetry, particularly in applications involving three-dimensional surface and/or volume change analysis.

## 2 Study site

The study basin (Figure 1), a tributary of the Leopold Burn located in the Pisa Range of the Southern Alps/Kā Tiritiri-o-te-Moana of New Zealand (44.882°S, 169.081°E), is 0.41 km$^2$ in size, and, and has been the subject of prior snow-hydrology investigations (Sims and Orwin, 2011). Elevation of the basin ranges between 1440 and 1580 m a.s.l. with a near-uniform area-elevation distribution (Figure 1). Average slope is moderate, with 80% of the basin having a surface slope of 24° or less. The basin runs north to south, and is drained by a small stream. While east of the Main Divide of the Southern Alps, the Pisa Range is representative of several large fault-block mountain ranges that dominate the eastern portion of the Clutha Catchment within the Otago region. These ranges are bounded by moderately steep slopes, rising to broad continuous ridge and plateau systems, in turn dissected by relatively shallow gullies, basins and gorges. These ranges feature relatively large areas above the winter snowline, with complex micro-terrain features, which are of interest in the context of re-distribution, preferential accumulation, and ablation of seasonal snow. In combination with typically windy conditions, the topography is expected to produce complex, highly variable spatial distributions of seasonal snow, convolved with, and potentially overtaking the role



of elevation in influencing variability in snow depth. The basin mapped in this study is larger than areas mapped for other similar studies published to date (Vander Jagt et al., 2015; Bühler et al., 2016; De Michele et al., 2016; Harder et al., 2016), and has a relatively complex topography, with several gullies dissecting the main slopes, separated by broad, steep sided ridges.

A visual assessment of Landsat, Sentinel-2, and MODIS imagery revealed that while the basin could be considered to be
in the marginal snow zone, snow cover persists from June to late September most years, thus providing opportunities for repeated mapping and the capture of the snowpack in various states.

# 3 Data and methods

## 3.1 Field approach

### 3.1.1 RPAS platform and payload

We used the Trimble UX5 Unmanned Aircraft System, a fixed wing RPAS manufactured by Trimble Navigation for photogrammetric applications. A single two-blade propeller, driven by a 700 W electric motor, propels the platform. Power is supplied from a 14.8 V, 6000 mAh Lithium-polymer battery allowing a flight endurance of 50 minutes. Autonomous navigation is supported by a single channel GPS receiver, which also provides approximate coordinates for each photo centre, while an accelerometer logs orientation data.

Imagery is captured by a large-sensor (APS-C) Sony NEX 5R mirrorless reflex digital camera providing a maximum imaging resolution of 4912 pixels by 3264 pixels, or about 4 cm GSD at 400 ft a.g.l. The camera is fitted with a Voigtländer Super Wide-Heliar 15 mm f/4.5 Aspherical II lens, with focus fixed to infinity. Appropriate exposure to ensure suitable contrast on the range of imaged targets was achieved with maximum aperture, high shutter speeds between 1/2500-1/4000 sec to minimize forward motion blurring, and automatic ISO sensitivity. Camera settings were checked prior to each flight to
accommodate varying light conditions and the relative share of ground cover (vegetation vs. snow).

### 3.1.2 RPAS flights

Three RPAS missions were undertaken with identical planning and differing states of snow cover in the basin (Table 1). All flights imaged 15 strips, aligned along the major axis of study basin (Figure 2). The study area was imaged with 90/80% forward/sideward overlap with respect to the lowest elevation to ensure that sufficient overlap was maintained when mapping
rising ground. The duration of each flight was ~35 minutes, with about 900 images being captured per flight.

### 3.1.3 Ground control survey

Achieving a robust constraint of exterior orientation parameters during aero-triangulation (AT) depends on the availability of a set of high-quality ground control points (GCPs). This is particularly true where the imaging platform lacks precise point positioning (PPP) capability (e.g., it carries only single frequency GPS and is not capable of determining differentially



corrected positions). Such code-only GPS navigation is accompanied by uncertainties two orders of magnitude greater than the expected accuracy of the models. Ground control networks were established for each RPAS flight mission using real time kinematic (RTK) Global Navigation Satellite System (GNSS) surveying with a Trimble R7 base station and R6 rover units. GCP locations were measured with accuracy on the order of ~2-3 cm. GCPs were signalled with $0.6 \times 0.6$ m mats painted

with a high contrast circular quadrant pattern for the autumn and winter flights. For the spring flight, chalk powder was used with a stencil to mark the target directly on the snow surface, using the same pattern as for previous flights. The use of chalk powder eliminated the need to retrieve GCP targets following RPAS flights. All survey work, as well as production of deliverables from photogrammetry, was carried out in terms of the New Zealand Transverse Mercator (NZTM) reference system.

It is well established that photogrammetric control is best achieved within the bounds of the GCP network (Linder, 2016), while the uncertainty associated with the geo-location of resected points increases beyond the control network. To constrain the area within the study basin for photogrammetric processing, the GCP network was distributed around the basin perimeter, as well as through the central area of the basin. Additionally, placement of GCPs on the valley floor and at mid-elevation within the basin ensured that the network also sampled the elevation range of the basin. An extensive GCP network was

established for the first flight with no snow on the ground, which permitted the robustness of AT to be tested under different GCP scenarios, as discussed further in Section 3.2.1. This allowed the network to be refined and reduced in size for subsequent missions, a matter of practical importance when working in alpine areas during the winter. Control point networks for each mission are illustrated in Figure 2.

### 3.1.4 *In situ* snow depth measurements

To assess the quality of snow depth data derived from RPAS photogrammetry, independent measurements were collected by manual snow probing on 10/09/2016, the same day as the spring RPAS mission. The sampling strategy involved the measurement of snow depth every 50 m along three elevation contours within the study basin, namely 1460, 1500 and 1540 m (Figure 2). This strategy ensured that snow depth was measured across a representative sample of slope aspect and elevation, while optimizing navigation across the basin. Snow depths were measured at each location by probing five times within arm

reach, and the location of the central measurement surveyed with RTK GNSS. This provided 430 measurements of snow depth, with the mean snow depth at each of the 86 locations providing a sample for comparisons with RPAS-derived snow depth.

### 3.2 Data processing

### 3.2.1 Photogrammetric processing

The goal of photogrammetry is to transform a set of images into a scene in which geometrically accurate measurements can

be made in three dimensional, often geographic, space. This requires a transformation from the inherent coordinate system of the device capturing imagery (a camera) to an appropriate geographic coordinate system (Vander Jagt et al., 2015; Linder,



2016). While traditional photogrammetry has long relied on metric (calibrated) cameras, the use of off-the-shelf non-metric cameras requires the simultaneous solving of both interior orientation (the camera model) and exterior orientation. This process, known as self-calibration, applies a bundle-block adjustment to solve the camera model describing the precise focal length ($f$), the offset between the principal point of autocollimation (PPA) and the centre of the imaging sensor plane ($x_0, y_0$), and the departure between the image point coordinate ($x,y$) and the idealized linear projection due to lens distortion. Camera calibration parameterises radial and decentering distortion with models such as that of Brown 1971:

$$\overline{x}' = (1 + K_1 r^3 + K_2 r^5 + K_3 r^7)\overline{x} + 2T_1\overline{xy} + T_2(r^2 + 2\overline{x}^2)$$
$$\overline{y}' = (1 + K_1 r^3 + K_2 r^5 + K_3 r^7)\overline{y} + 2T_2\overline{xy} + T_1(r^2 + 2\overline{y}^2)' \tag{1}$$

in which

$$\overline{x} = x - x_0, \tag{2}$$
$$\overline{y} = y - y_0, \tag{3}$$
$$r = \sqrt{\overline{x}^2 + \overline{y}^2}. \tag{4}$$

Image coordinates corrected for lens distortion are then used in the set of collinearity equations relating object point coordinates ($X_A, Y_A, Z_A$) to the corresponding image point coordinates ($\overline{x_A}', \overline{y_A}'$) to solve for the exterior orientation (Vander Jagt et al., 2015; Linder, 2016):

$$\begin{pmatrix} \overline{x_A}' \\ \overline{y_A}' \end{pmatrix} = \begin{pmatrix} f\frac{r_{11}(X_A-X_0)+r_{12}(Y_A-Y_0)+r_{13}(Z_A-Z_0)}{r_{31}(X_A-X_0)+r_{32}(Y_A-Y_0)+r_{33}(Z_A-Z_0)} \\ f\frac{r_{21}(X_A-X_0)+r_{22}(Y_A-Y_0)+r_{23}(Z_A-Z_0)}{r_{31}(X_A-X_0)+r_{32}(Y_A-Y_0)+r_{33}(Z_A-Z_0)} \end{pmatrix}. \tag{5}$$

The $r_{ij}$ terms represent the 3×3 rotation matrix relating the sensor coordinate system orientation to the ground coordinate system. Since the UX5 camera is fixed with respect to the platform, the latter combines the roll, pitch and yaw ($\omega, \varphi, \kappa$) of the platform at the time of exposure.

### 3.2.2 Software

Initially, aero-triangulation was carried out using the photogrammetry module of Trimble Business Center, v3.40 (TBC), which relies on an implementation of the adjustment process from Inpho UAS Master. Deliverables produced using TBC, however, suffered from severe elevation artefacts which limited their usefulness for further analysis. These pitfalls are discussed further in Section 5.3.

Subsequent to the identification of shortfalls in TBC, aero-triangulation was carried out using Trimble Inpho UAS Master® v8.0. The solution is initialised by the positional parameters ($X_0, Y_0, Z_0$) for each photo centre, as provided by the on-board GPS receiver. Relative adjustment is achieved after automatic collection of tie points (TPs). Subsequent measurement of GCPs allows absolute adjustment that refines the exterior orientation, as well as solves for the interior orientation parameters.

The robustness of photogrammetric modelling was assessed via evaluation of several alternate control scenarios, based on the first mission when 23 GCPs were placed and measured in the field. This assessment aided the determination of an



optimal number of GCPs to minimise the time required to place and survey control points when snow is present in the basin. The following scenarios were evaluated:

1. All 23 control points as horizontal and vertical GCPs
2. 14 control points as horizontal and vertical GCPs
3. 6 control points as horizontal and vertical GCPs

In each scenario, the balance of the control points was provided as check points (CP), and as the number of GCPs decreased, the magnitude of the Root Mean Square Error (RMSE) for CPs was used to assess triangulation robustness. It was found that as few as 14 GCPs provided an acceptable triangulation across the study area, with some degradation apparent when only 6 GCPs were used, primarily in terms of z (Table 2). No spatial structure was evident in the distribution of GCP or CP error. On this basis, 14 control points were placed and measured in the field for each of the winter and spring missions, with eight of those set as horizontal and vertical GCPs, and the remaining six as CPs.

### 3.2.3 Deliverables

Standard deliverables from the photogrammetric modelling included a dense point cloud; a digital surface model, interpolated to 0.15 m spatial resolution; and an ortho-mosaic, resampled to 0.05 m spatial resolution. The digital surface model (DSM) and the ortho-mosaic are the principal products for further analysis. The DSM for each epoch provides the basis for determining snow depth, while the ortho-mosaics allow for assessment of the snow-covered area, and for snow-free areas to be identified when assessing the quality and repeatability of DSMs between flight missions. Deliverables were generated from photogrammetric models utilising all surveyed points as GCPs in the aero-triangulation. A second aero-triangulation was then run using a subset of control points as CPs based on scenario 2 from Table 2. Thus, the RMSE provided from CPs is expected to be conservative compared to the quality of the deliverables obtained from the fully constrained AT.

### 3.2.4 Derivation of snow depth

Snow depth was derived by differencing DSM of flights 2 and 3 from the baseline obtained during flight 1 (*ref*) as:

$$dDSM_n = DSM_n - DSM_{ref}. \tag{6}$$

Equation 6 provides a map of difference between the two DSMs, henceforth referred to as the *dDSM* (after Nolan et al., 2015). Values of the dDSM are considered to represent snow depth, with associated uncertainty considered in Section 3.3.

### 3.3 Quality and accuracy assessment

Summary statistics, typically based on the RMS error of GCPs and CPs from the aero-triangulation, indicate the expected accuracy of deliverables. Since snow depth is determined by differencing two DSMs, associated uncertainty should be determined via error propagation. The overall accuracy of the DSM differencing approach, should also be validated against reference data (e.g., snow depth measured *in situ*), temporally coincident with RPAS measurements. Areas of snow-free terrain



during Flight 3 further supplement snow depth observations by providing an extensive source of samples to assess the repeatability of the photogrammetric modelling process.

Previous studies have considered the accuracy of RPAS-derived snow depth by comparison with reference data from *in situ* snow depth alone (Bühler et al., 2016; De Michele et al., 2016; Harder et al., 2016), while ignoring the uncertainty inherent
to each photogrammetric model and their propagation into the dDSM. Here, the accuracy of photogrammetrically derived snow depth is assessed by exploring both approaches. Relating photogrammetric model quality, as inferred from GCPs/CPs, to observed uncertainties in the determination of snow depth provides the basis for realistically informing uncertainties in snow depth from ongoing RPAS measurements. This in turn allows rigorous inferences about the evolution of snow depth to be made, without the need for further campaigns of *in situ* validation.

**3.3.1 Uncertainty associated with RPAS-derived snow depth**

Since snow depth is determined via DSM differencing as a linear combination of two independently measured variables (Equation 6), the uncertainty associated with snow depth (SD), measured in the vertical dimension, for each epoch ($n$) can be obtained via Gaussian error propagation (James et al., 2012) as:

$$\varepsilon_{dDSM} = \sqrt{\varepsilon_n^2 + \varepsilon_{ref}^2}, \tag{7}$$

where ε for each DSM is the elevation error determined from the aero-triangulation as the $RMSE_Z$ value for the set of CPs. This simple approach assumes that the planimetric precision of each constituent *DSM* has negligible contribution to $\varepsilon_{dDSM}$. Calculating $\varepsilon_{dDSM}$ provides a single estimate of uncertainty assumed to apply equally throughout the map of RPAS-derived snow depth for each epoch. Under the assumption that errors are normally distributed and bias-free, the $RMSE_z$ derived from CPs identifies to standard deviation $\sigma_z$, allowing the 90% confidence level of $z$ to be determined as $1.65 \times \sigma_Z$. In turn, inferences
associated with uncertainties for elevation differences, $\varepsilon_{dDSM}$, also depend on the Gaussian assumption to provide the 90% confidence level.

In reality, perfect co-registration between constituent DSMs, and the Gaussian assumption, are unwarranted. Subsequently, inferences associated with the evolution of snow depth may be compromised due to confidence intervals being conservative or immoderate. Therefore, we use *dDSM* for snow-free areas to characterise the experimental distribution of
25 errors and assess the validity of the Gaussian assumption in this context.

**3.3.2 Validation against reference *in situ* snow depth measurements**

The approach above provides a means to determine the expected accuracy of snow depth derived from RPAS photogrammetric surveys. In order to validate this estimate, a reference dataset of *in situ* observations was sampled in the field using snow probes, with a nominal accuracy of ± 1 cm, as described in Section 3.1.4. De Michele et al (2016) assessed the accuracy of
30 RPAS-derived snow depths against snow depth surfaces interpolated from 12 point measurements. This approach, however,





may be limited by an inability to accurately resolve the spatial variability of snow depth, as well as the compounding effects of uncertainty associated with the interpolation scheme, particularly beyond the domain defined by the measured points.

Here, 430 measurements of snow depth provided 86 mean reference values, with the standard deviation of each set of five measurements providing 95% confidence intervals. The aim of this sampling strategy was to assess and account for co-location

uncertainty and spatial variability between the RPAS and reference snow depth datasets. Reference snow depths were compared with those from the spatially coincident pixels from the map of RPAS-derived snow depth. RPAS-derived snow depth quality was assessed in terms of residuals and weighted linear regression between reference and RPAS-derived snow depths.

### 3.3.3 Repeatability of photogrammetric modelling

Emergence of snow-free areas at the time of the spring flight facilitated comparison between autumn and spring DSMs on those areas. As the same terrain surface mapped from two independent flights should yield identical DSMs, the residual between them provides a means to characterise the distribution of errors in the photogrammetric processing, which can be readily compared to the assessment made from CPs. Ultimately, this residual represents a measure of the repeatability of the technique for measuring surface height change.

Snow-covered and snow-free areas were segmented using an unsupervised classification of the ortho-image, able to discriminate between illuminated snow pixels, shaded snow pixels, and vegetation and soil dominated snow-free pixels. Snow-free pixel classes were then grouped to provide a mask within which the distribution of $dDSM_3$ values could be characterized.

### 3.3.4 Resolution of fine scale spatial variability

A primary motivation for exploring the use of RPAS photogrammetry for mapping a snow pack is the ability to resolve fine

scale spatial variability in snow depth. This capability was assessed by computing and comparing the semi-variograms of reference and RPAS-derived snow depths from the autumn flight. While the sample size for reference snow depths remained fixed ($n$=86), the semi-variogram of RPAS-derived snow depths could be calculated from many more samples. Two random samples were extracted from the $dDSM_3$ map ($n = 1000$ and $n = 5000$), each yielding a semi-variogram capturing the spatial variability of snow depth with increasing detail, which were compared to that of the *in situ* observations.



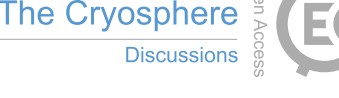

## 4 Results

### 4.1 Photogrammetric processing

#### 4.1.1 Quality of the triangulation

**RMSE for GCPs for all flights was at the centimetre to sub-centimetre level (**

Table 3). Since GCPs are used to solve the photogrammetric model, they do not provide an independent assessment of accuracy. Such an assessment is provided by the CPs, the RMSE of which was on the order of centimetres for all flights. Planimetric RMSE (i.e., $x$ and $y$) was always substantially less than the GSD. Vertical RMSE ($z$) tended to be about double that achieved planimetrically, but never exceeded 5 cm. The final models were produced from a second and more constrained AT with all surveyed points used as GCPs, thus making the assessment conservative relative to the final products.

While the RMSE of CPs increased for the winter and spring epochs, possibly due to a less constrained model, the level of accuracy achieved is compatible with expectations for the determination of snow depth. Additionally, the more tightly constrained first epoch reduced the error for the baseline model, in turn contributing a reduced uncertainty in derived snow depths, despite the reduced control for subsequent epochs. For all flight missions, the photogrammetric processing performed well in the correlation of images and the construction of the image block, as indicated in Table 3. Tie point (TP) generation

relies on the successful match of unique targets across multiple images, which was achieved despite the complicated contrast over snow. For all flights >80,000 TPs were generated across the imaged area.

#### 4.1.2 Determination of snow depth

Snow depth was found to be highly variable across the study basin for both epochs two and three (Figure 3). The mid-winter epoch exhibited near complete snow cover across the study basin, while large snow-free areas developed in the spring epoch,

where snow covered area was reduced by about one third (Figure 3 A & B). Where snow was present, depths ranged from a few centimetres, typically on exposed ridgelines and broad elevated slopes, to two metres or more where cornices formed along ridgelines, as well as in gullies. Average snow depth was greater at the winter epoch, although maximum depths were comparable between both epochs. Between winter and spring, considerable ablation was observed. Areas of deepest snow were spatially coincident between winter and spring, with the greatest retention of snow in cornices and gullies. Where shallow

snow was present on ridgelines in winter, it was largely lost by spring.

### 4.2 Accuracy assessment and validation of snow depth

#### 4.2.1 Propagation of aero-triangulation error

Propagation of errors under the Gaussian assumption, based on the RMSE from each aero-triangulation, yielded vertical uncertainties for snow depths at the 90% confidence level of ±0.077 m for the winter flight and ±0.084 m for the spring flight.



This one-dimensional approach to error propagation assumes that the planimetric geolocation of individual surfaces, and subsequently the co-registration of surface pairs does not contribute significantly to the vertical uncertainty.

### 4.2.2 Assessment against reference probe data

Comparison of RPAS-derived and reference snow depth yielded a mean residual of -6.9 cm, indicating that, on average, reference depths were greater than RPAS-derived depths. Filtering the reference dataset to exclude reference measurements that were made in areas occupied by tussock (*Chionochloa rigida*) vegetation, however, improved the mean residual to -1 cm (Figure 4A). The small residual is indicative of good agreement between the two datasets, while also indicating that overall, snow depths measured by probing may be overestimated. Limitations of probing and uncertainty introduced due to the presence of vegetation is discussed further in Section 5.2.1.

Good agreement between both datasets is further demonstrated in Figure 4B. Relatively large horizontal error bars accompanying the reference measurements (Figure 4B) reflect the substantial spatial variability in snow depth measured by probing, even within arm's reach. Substantial departure occurs for reference snow depths between 20 and 60 cm which tend to exceed RPAS measurements. Negative depths in the RPAS-derived dataset is a product of co-registration uncertainty, particularly in areas where the surface model represents large vegetation, or is influenced by rock outcrops, as well as spurious values from the constituent DSMs. Agreement between reference and RPAS-derived datasets improved with the removal of reference measurements made above of tussocks. This filtering saw the $R^2$ value improve by 22%, while RMSE decreased by 23% (Table 4). The 1:1 ratio line was contained within the 95% confidence interval of the weighted (bisquare) regression between RPAS-derived and filtered reference snow depths.

### 4.2.3 Comparison of DSMs from independent RPAS flights

The emergence of snow-free areas for the September flight permitted a comparison of height derived on snow-free surfaces between the pre-winter and spring flights (Figure 5). The small magnitude of the residuals, compatible with errors consistent with the uncertainty of the triangulation CPs, demonstrates the repeatability in the derivation of snow-free surfaces. Furthermore, the absence of any spatially structured trend in the distribution of the residual indicates robust photogrammetric modelling from the RPAS platform. At 0.15 m resolution, the snow-free pixels from the spring mission provided a large sample ($n = 5936428$). The mean residual (bias) detected with respect to the pre-winter DSM was 0.024 m ($\sigma = 0.239$ m) (Figure 6).

The set of residuals departed substantially from the Gaussian distribution, and was better represented by the Student's *t* location-scale distribution (Figure 6):

$$f(x) = \frac{\Gamma\left(\frac{v+1}{2}\right)}{\sigma\sqrt{v\pi}\Gamma\left(\frac{v}{2}\right)}\left(\frac{v+\left(\frac{x-\mu}{\sigma}\right)}{v}\right)^{-\left(\frac{v+1}{2}\right)}, \qquad (9)$$

where $\mu$, $\sigma$ and $v$ are the location, scale and shape parameters, respectively. Large kurtosis (calculated $k = 1956$) associated with the histogram of residuals in Figure 6 shows significant departure from a Gaussian law (for which $k = 3$) of equal standard




deviation, $\sigma$. The leptokurtic experimental distribution results in a narrower 90% confidence interval than that estimated under the Gaussian assumption with $\sigma = 24$ cm, while the probability of large residuals is larger than predicted by a Gaussian distribution. Overall, the mean residual ($\mu = 2$ cm) and the precision of ±13 cm (90% confidence level, calculated from the distribution 90th percentile) exceeds the uncertainties estimated from error propagation alone (±8 cm at 90% confidence level,

see Section 4.1.1), yet support the suitable repeatability of the photogrammetric modelling. Importantly, the significant departure from a normal distribution shows that assessing the variability from a Gaussian fit on stable targets (±39 cm at the 90% level) would significantly overestimate the confidence interval. On the other hand, the 90% confidence interval calculated from the fitted Student's $t$ location-scale is ±10 cm (Table 5). The significance of this result with respect to statistical inferences is discussed further in Section 5.2.2.

The non-Gaussian nature of the residual distribution deserves further scrutiny. Similar distributions have been identified for comparable repeatability assessments of photogrammetric dDSMs used for mapping snow depth (Nolan et al., 2015), but have not been explored in detail. Analysing the variability of the mean and standard deviation of the residual, as well as the kurtosis of the residual distribution, for discrete classes of slope, provided insight into the role of terrain. For classes of slope up to 65° the mean residual remains within the standard error, before becoming increasingly negative for the remaining classes

(Figure 7). Standard deviation exhibits a similar trend, remaining largely within the overall standard error for slope classes up to 45°, beyond which variability increases.

The observed pattern in the mean and standard deviation of the residual indicates that larger and more variable errors are associated with steeper slopes. Reduced kurtosis accompanying the error distribution on larger slopes (Figure 7) reveals a tendency towards a Gaussian distribution of residuals as mean slope increases. Here, for slopes >50°, kurtosis was reduced

below 100, and for slopes >85°, kurtosis was less than 10, approaching that of the normal distribution. Therefore, the statistical distribution of error, whilst non-normal, also varies significantly with terrain characteristics (Figure 8). Subsequently, the overall distribution of residuals (Figure 6) is the result of a convolution between non-normal distributions and the hypsometry of the area (i.e., area-elevation distribution).

### 4.2.4 Characterising the spatial variability of snow depth

The semi-variograms for RPAS-derived snow depth, compared to that from the reference measurements, are shown in Figure 9. They exemplify the new insight that high-resolution mapping provides into the spatial variability of snow depth. Both the 1000 and 5000 random point samples captured a comparable structure of spatial auto-correlation with a range of ca. 40 m. The 5000-point sample improved the resolution of the semi-variogram, with an improved signal to noise ratio. In contrast, the reference data, despite being demanding in fieldwork, performed poorly at capturing the spatial variability, as most

measurements were separated by a minimum distance of 50 m. A lack of spatial auto-correlation in the reference data confirms a-posteriori that probing samples could be assumed to be independent of each other, which is desirable for the accuracy assessment. Additionally, it also reveals that probing failed to capture most of the spatial structure of the snow depth field, thus stressing a limitation of this classical method to characterise the snowpack.





## 5 Discussion

### 5.1 Performance of RPAS photogrammetry for resolving snow depth

Overall, RPAS photogrammetry has been found to be suitable for determining snow depth via DSM differencing. Primarily, the achievement of uncertainties <0.13 m at the 90% confidence level for derived snow depth provides a basis for useful data capture, and robust inferences and interpretations. The reported magnitude of uncertainties account for the sources discussed further below, and compare favourably with other similar studies (Vander Jagt et al., 2015; Bühler et al., 2016; De Michele et al., 2016; Harder et al., 2016). Decimetre levels of uncertainty appear to be an emerging benchmark for snow depths measured by RPAS photogrammetry, and also considered as standard for airborne LiDAR (Deems et al., 2013). In terms of comparisons with *in situ* data, Figure 4 shows good agreement between RPAS and reference snow depth, and that RPAS photogrammetry performance improves as snow depth increases. At the same time, use of probed snow depths as references for validating such data can be compromised by the nature of the underlying vegetation.

Mapping snow depth continuously at 0.15 m resolution, across an entire hydrological basin, represents a new contribution to the quantification and characterisation of spatial variability in snow depth at this scale. Before considering the broader implications of this in terms of snow processes, uncertainty, limitations, and pitfalls of the approach are considered.

### 5.2 Sources and nature of uncertainty

#### 5.2.1 Vegetation

Vegetation contributes to uncertainty, particularly when validating RPAS-derived snow depths against reference snow depths. As described in Section 4.2.2, the agreement between RPAS-derived and probed snow depths improved substantially when considered only away from large tussock vegetation. It is likely that the presence of tussock introduces a bias into the snow depth measurement, whereby a probe may penetrate the tussock foliage, and possibly also a sub-vegetation void, before striking the ground surface. Similar challenges have been documented by Vander Jagt et al. (2015). High resolution dDSMs, on the other hand, resolve the vegetation surface, and so vegetation height is inherently accounted for.

As identified by Nolan et al. (2015), photogrammetrically-derived snow depths may also be affected by the compaction of vegetation below the snowpack, which may introduce an anomalous signal of surface height change, to the point of returning false negative snow depths. Correcting observed surface height change would not be straightforward, and is not possible with the data acquired within this study. The effects of vegetation compaction are likely to be greatest in the early winter. As grass typically does not rebound until after the complete removal of the winter snowpack, ongoing subsidence of vegetation below the snowpack through mid-winter and spring is expected to be minimal.

Ultimately, this study suggests that for areas dominated by tussock vegetation, RPAS photogrammetry may provide a more reliable means of measurement than probing. A lack of knowledge regarding the specific location of sub-snow vegetation when making measurements by probing is likely to provide a systematic over-estimation of snow depth (Figure 4).




### 5.2.2 Geo-location and co-registration

The assumption that error distribution associated with physical measurements is normal often underpins subsequent statistical inferences. As demonstrated in Section 4.2.3, the error associated with the bias between independently acquired DSMs significantly departed from normal, and was better approximated by the Students $t$ location-scale distribution. This extremely

leptokurtic distribution of residuals reflects the influence of relatively low frequency, but high magnitude residuals beyond the probability of the normal law, despite an overall dominance of residuals about and close to the mean. A possible source of large residuals between two DSMs is their relative planimetric accuracy, and subsequent co-registration quality (Kääb, 2005). For steep terrain in particular, a horizontal displacement between DSMs could add a component to dDSM uncertainty beyond the vertical accuracy of constituent DSMs. The residual ($\Delta h$) between two surface profiles, which are identical but horizontally

displaced by 0.5 m, is shown in Figure 10A. The error introduced to DSM differencing resulting from co-registration uncertainty increases with steepening slope. Maximum residuals coincide with the steepest terrain (near vertical areas associated with rock outcrops), and exceed two metres. The sign of the error is aspect dependent, assuming a uniform horizontal displacement.

        Consistent with Kääb (2005), the vertical error introduced by a uniform, one-dimensional (e.g., horizontal) offset, is given

by:

$$\Delta h = dx \tan \theta, \tag{9}$$

where $dx$ is the offset between transects (i.e., 0.5 m in this case), and $\theta$ is the surface slope in degrees, as seen in Figure 10B. It is clear from Figure 10B that where the average slope of target surfaces is low and co-registration quality is good, the error introduced to a dDSM as a product of co-registration will be minimal. Increasing slope and/or co-registration uncertainty is

accompanied by increased vertical uncertainty in the dDSM. This relationship is consistent with the findings of Section 4.2.3 resulting in the distribution of residuals departing substantially from the Gaussian distribution when the proportion of steep slopes is low. Complicating this effect is the fact that co-registration uncertainty exists in two dimensions. Subsequently, it will become dependent on aspect as well as slope (Nuth and Kääb, 2011), with neither possessing a uniform spatial distribution. This effect is expected to be more pronounced with very high resolution (i.e., sub-metre) surface models due to a greater

frequency and magnitude of breaks in surface slope being resolved compared to coarser models. The modification of surface slope for constituent DSMs (e.g., through the addition of snow) further convolves the propagation of vertical uncertainty.

        Despite this, the leptokurtic observed error distribution indicates that the reliance on statistics that assume a Gaussian distribution of errors will provide an over-estimated characterisation of the expected accuracy. Over-estimation of uncertainties may in turn affect statistical inferences and the computation of uncertainties on derived parameters.

The convolution of vertical and planimetric accuracy stresses the importance of ensuring a robust aero-triangulations and the benefits of utilising high quality ground control. With new photogrammetric platforms leveraging non-metric cameras and resulting image blocks prone to sub-optimal photogrammetric modelling (Sirguey et al., 2016), there is a need to be wary of systematic bias, or spatial structure, in the distribution of errors, which may not be revealed readily by residuals from the aero-



triangulation alone. These considerations are especially important where a relatively high level of precision is required, and the signal to noise ratio may be low, when assessing relatively subtle surface height and/or volume changes from dDSMs. Utilising independent aero-triangulations as the control of co-registration quality, rather than explicitly co-registering DSMs, has the further advantages of simplifying the processing chain from data acquisition to change detection, mitigating against

the risk of introducing gross error when co-registering DSMs and avoiding the need for snow-free (or stable) reference areas within the analysis region.

### 5.3 Pitfalls and limitations of RPAS photogrammetry

Initial processing, using the photogrammetry module of Trimble Business Center (v3.40) produced strong striping artefacts in the dDSM. Striping involved a periodic bias in surface height change, aligned with the 15 image strips. This was readily

revealed due to identical flight plans between successive epochs making constructive errors obvious, rather than convoluted with terrain variability. This systematic error was severe and problematic, particularly when considering the surface change resulting from the addition of snow cover to the ground. Changing surface height concealed stable references, precluding characterisation of the error and its empirical removal from the real signal of surface height change (e.g., Albani and Klinkenberg, 2003; Berthier et al., 2007). Products derived using UAS Master (v8.0) did not exhibit such artefacts, allowing

the potential source of the systematic error to be explored, and highlighting potential pitfalls in RPAS photogrammetry.

The lack of systematic bias in dDSMs derived from processing in UAS Master indicates that the latter provides a more reliable aero-triangulation. Thus, the UAS Master triangulation provided a reference surface for further exploration of the nature of the bias propagated in the TBC triangulation. Comparisons are provided from the winter flight here. Given the nature of the photogrammetric problem, small errors in either or both interior orientation, as described by the camera calibration, or

the rotational components of the exterior orientation (i.e., roll, pitch, yaw; $\omega$, $\varphi$, $\kappa$, respectively), can result in large errors in the adjusted image block with a spatially structured pattern (Sirg*uey* et al., 2016).

Firstly, interior orientation was assessed by comparing the distortion models provided for each of the two software calibrations of the same camera, for the same flight. No significant difference was detected (Figure 12), with only a small divergence in lens distortion occurring at >10 mm radius, reaching 1% at 14.5 mm. The observed agreement between lens

distortion models indicated that a difference in the interior orientation solution between TBC and UAS Master was not the source of the artefacts seen in products of the TBC triangulation.

Since the observed artefact was propagated along the flight lines, the nature of the roll ($\omega$) was considered. Bias in the estimation of this parameter could lead to a systematic elevation offset of resected points between flight lines, either raising or lowering the terrain alternatively, as documented in the case of stereo-satellite imagery by Berthier et al. (2007). Occurrence

of this for multiple flights with near identical flight lines would exacerbate the constructive biases, resulting in the striping in the dDSM. Alternatively, pitch and yaw parameters are unlikely to produce such an artefact along the flight direction. The residual of $\omega$ for individual photo centres between each of the two software packages confirmed a positive bias in the $\omega$ value



as estimated by TBC v3.40 relative to that provided by UAS Master (Figure 13). The mean residual ($\overline{r\theta}$) was found to be 0.014°.

The impact of bias in ω on the resected height $h$ for a target with respect to a photo centre can be estimated simply as a right-angle triangle, since values of ω are small compared to the baseline length, $l$, which is equal to half the distance between adjacent flight lines (see Figure 14):

$$\tan\theta = \frac{h}{l}, \tag{10}$$

$$\tan(\theta - \overline{r\theta}) = \frac{h - \Delta h}{l}, \tag{11}$$

$$\Delta h = h - l\tan(\overline{r\theta}), \tag{12}$$

$$\Delta h = h - l\tan\left(\arctan\frac{h}{l}\ \overline{r\theta}\right). \tag{13}$$

Using the observed value of 0.014° for $\overline{r\theta}$, $\Delta h$ was calculated for a range of typical values of $h$, yielding the relationship between $h$ and $\Delta h$ as shown in Figure 15. Bias in the estimation of the ω parameter during the aero-triangulation has the potential to introduce a significant vertical error, dependent (non-linearly) on flying height ($h$), propagating an error of ±0.12 m at a flying height of 110 m. The increase in error with flying height above ground level was consistent with the observed propagation of striping artefacts in DSM products, whereby the magnitude of the observed bias decreased as terrain height increased (Figure 11), while absolute flying height remained approximately constant.

The observed propagation reinforces the need for vigilance when working with such datasets, particularly those delivered from "off the shelf" photogrammetry packages, which are becoming increasingly popular. Artefacts such as the striping identified here, and evidence of non-optimal aero-triangulation, are likely to be less obvious as the complexity of the mapped terrain increases.

**5.4 Spatial and temporal trends in snow cover**

Figure 9 demonstrates the new insight that RPAS photogrammetry can provide over probing for resolving spatial variability in snow depth, particularly at fine scales. Therefore, RPAS photogrammetry can provide a basis for improving spatially distributed snowpack models. In turn, this contribution will further improve understanding of seasonal snow processes, including redistribution, preferential accumulation and ablation. Such data can facilitate the use of geostatistical approaches for examining controls on spatial distribution of snow, such as that applied to the Brewster Glacier, New Zealand (Cullen et al., 2017). In this case, the density of measurements provides insights into spatial variability at scales that would allow consideration of terrain and meteorological controls on snow distribution at micro-scales, extending understanding beyond the spatial co-variance between snow depth and elevation.

The ability to resolve fine scale variability reliably from continuous raster snow maps lessens the dependence on interpolation through areas of sparse data for interpreting controls on spatial distribution of snow. While previous studies have been able to correlate between snow and terrain properties (e.g., Anderson et al., 2014), such studies rely on the inference of basin scale processes from transect scale observations. The ability to produce spatially continuous maps of snow depth, across





an entire basin, at a resolution of 0.15 m bridges this gap and reduces the reliance on inferences when scaling up from point- or transect-based *in situ* observations to basin scale processes. Such datasets provide an opportunity to build on previous work in understanding the relationships between snow re-distribution, preferential accumulation and ablation, terrain and meteorology (Winstral et al., 2002, 2013; Webster et al., 2015; Revuelto et al., 2016). Furthermore, resolving snow depth in

this way across an entire basin facilitates robust integration into hydrological models, enhanced by validation against catchment discharge (e.g., from stream flow data).

     The mapping of snow depth effectively provides a volumetric view of the snow pack across the basin (i.e., depth × area), the snow pack mass balance in terms of SWE can be calculated based on *in situ* measurements of snow density. While snow depth was only determined for two epochs in this case, emergent trends within the data can be explored. Between the winter

and spring flights, the basin snow covered area (SCA) decreased from 100%, to 67%. Simultaneously, SWE calculated using density measured from snow pits at both epochs, decreased by 20%. This highlights the importance of effective concentration of snow in preferred areas, and the complex spatial distribution that results. The ability to detect this, even with a temporally limited dataset, indicates the potential for RPAS photogrammetry as a measurement approach for improving resolution and understanding of snow hydrology. In particular, such datasets may offer a unique opportunity to assess the performance of

models forced by remotely sensed data of coarser resolution in estimating SWE from estimates of sub-pixel fractional SCA (Bair et al., 2016).

## 6 Conclusions and outlook

This study has demonstrated that RPAS photogrammetry provides a suitable, repeatable means of reliably determining snow depth in an alpine basin of low relief, but possessing some terrain complexity. Achieving decimetre level accuracy for

measuring snow depth provides a basis for monitoring of a seasonal snowpack and associated processes, especially considering the capacity to provide very high resolution, spatially continuous measurements across an entire hydrological basin. Such ability to characterise the seasonal snowpack will provide an important stepping-stone for improved modelling of seasonal snow and associated processes, especially through accurate mapping of an entire hydrological basin.

     Challenges encountered through this deployment provide important points for consideration in this and other applications

of close-range photogrammetry, especially from RPAS platforms, for surface and volume change analysis. Specifically, small but persistent bias in photogrammetric solutions for the roll parameter exemplify the possibility of sub-optimal solutions in processing software. Such bias can introduce substantial systematic errors which may be difficult to correct, and can compromise further analysis.

     We show that uncertainty analysis from the aero-triangulation only, and based on a limited number of check points, may

underestimate the uncertainty. Alternatively, an assessment of repeatability of photogrammetric modelling on stable ground can support a more detailed uncertainty analysis. It reveals, however, that the statistical distribution of the error of differentiated surface models is more complex than normal and governed by terrain parameters. The leptokurtic residual distribution



demonstrates that an assumption of Gaussian law can substantially overestimate confidence intervals, in turn compromising inferences. This result has important practical applications to the computation of uncertainties in studies that characterise volume change from repeated surface modelling.

Finally, there is scope to further refine the characterisation of uncertainty associated with RPAS photogrammetry in order to ensure that all potential sources of error are captured, and that statistical analysis is appropriate to the distributions within underlying data. Existing methods for mitigating the impact of co-registration uncertainty of coarser products may permit modelling and correction of such errors in the very high resolution products that are now available.

**Acknowledgments**

The authors are grateful to those who provided assistance in the field: Julien Boeuf, Kelly Gragg, Mike Denham, Craig MacDonell, Aubrey Miller, Sam West, Thomas Ibbotson and Alia Khan. Sean Fitzsimons provided helpful feedback on the draft manuscript. This research was carried out under Department of Conservation Research & Collection Authorisation 53609-GEO. This research was funded by a University of Otago Doctoral Scholarship with support from the Department of Geography and an internal grant of the National School of Surveying, as well as a Fieldwork Grant from the New Zealand Hydrological Society.

**Author contributions**

TR and PS designed the study and collected the data. TR processed and analysed the data and wrote the manuscript with input from PS and NC. PS and NC supervised the research.

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

| Mission/flight | Date | Season | Epoch | Snow cover |
| --- | --- | --- | --- | --- |
| M001f01 | 17/05/2016 | Autumn | 1 | Minimal - trace of early snowfall |
| M002f01 | 02/08/2016 | Mid-winter | 2 | Extensive – winter snowpack |
| M003f01 | 10/09/2016 | Spring | 3 | Spring melt underway, extensive snow-free areas |

**Table 2: Summary results of alternative GCP scenarios tested for aero-triangulation within UAS Master.**

| Scenario | GCP RMSE (m) | | | | CP RMSE (m) | | | |
| --- | --- | --- | --- | --- | --- | --- | --- | --- |
| | $n$ | $x$ | $y$ | $z$ | $n$ | $x$ | $y$ | $z$ |
| 1 | 23 | 0.0069 | 0.0076 | 0.0055 | 0 | N/A | N/A | N/A |
| 2 | 14 | 0.0017 | 0.0010 | 0.0004 | 9 | 0.0119 | 0.0184 | 0.0320 |
| 3 | 6 | 0.0033 | 0.0039 | 0.0009 | 17 | 0.0263 | 0.0207 | 0.0575 |

**Table 3: Summary statistics for each of the triangulations used to produce DSMs and ortho-mosaics from each of the three flight missions.**

| Flight | $n$ images captured | $n$ images used | $n$ TP | GCP RMSE (m) | | | | CP RMSE (m) | | | |
| --- | --- | --- | --- | --- | --- | --- | --- | --- | --- | --- | --- |
| | | | | $n$ | $x$ | $y$ | $z$ | $n$ | $x$ | $y$ | $z$ |
| 1 | 885 | 885 | 100,390 | 14 | 0.0083 | 0.0073 | 0.0034 | 9 | 0.0134 | 0.0163 | 0.0220 |
| 2 | 920 | 917 | 98,730 | 8 | 0.0067 | 0.0085 | 0.0018 | 6 | 0.0368 | 0.0293 | 0.0409 |
| 3 | 891 | 889 | 88,791 | 8 | 0.0105 | 0.0108 | 0.0028 | 6 | 0.0246 | 0.0247 | 0.0457 |



**Table 4: Parameters of weighted regression between reference and RPAS-derived snow depths.**

|  | $n$ | $\beta_0$ | $\beta_1$ | RMSE | $R^2$ | p-value |
|---|---|---|---|---|---|---|
| All points | 86 | 0.92 | 0.80 | 14.7 | 0.67 | 0.000 |
| Non-tussock | 52 | 1.69 | 0.86 | 11.3 | 0.82 | 0.000 |

**Table 5: Observed (calculated under Gaussian assumption) and fitted parameters for the residual distributions shown in Figure 6.**

| Parameter | Distribution | Value |
|---|---|---|
|  | Observed | 0.024 |
| $\mu$ | Normal fit | 0.036 |
|  | t l-s fit | 0.019 |
|  | Observed | 0.239 |
| $\sigma$ | Normal fit | 0.236 |
|  | t l-s fit | 0.056 |
| $\nu$ | t l-s fit | 2.579 |

10   **Table 6: Observed (calculated under Gaussian assumption) and fitted parameters for the residual distributions shown in Figure 8.**

| Parameter | Distribution | Slope class 5 - 10° | 70 - 75° |
|---|---|---|---|
|  | Observed | 0.026 | -0.022 |
| $\mu$ | Normal fit | 0.0257 | -0.0217 |
|  | t l-s fit | 0.0211 | -0.118 |
|  | Observed | 0.186 | 0.892 |
| $\sigma$ | Normal fit | 0.186 | 0.8922 |
|  | t l-s fit | 0.0463 | 0.3758 |
| $\nu$ | t l-s fit | 4.1037 | 2.0927 |





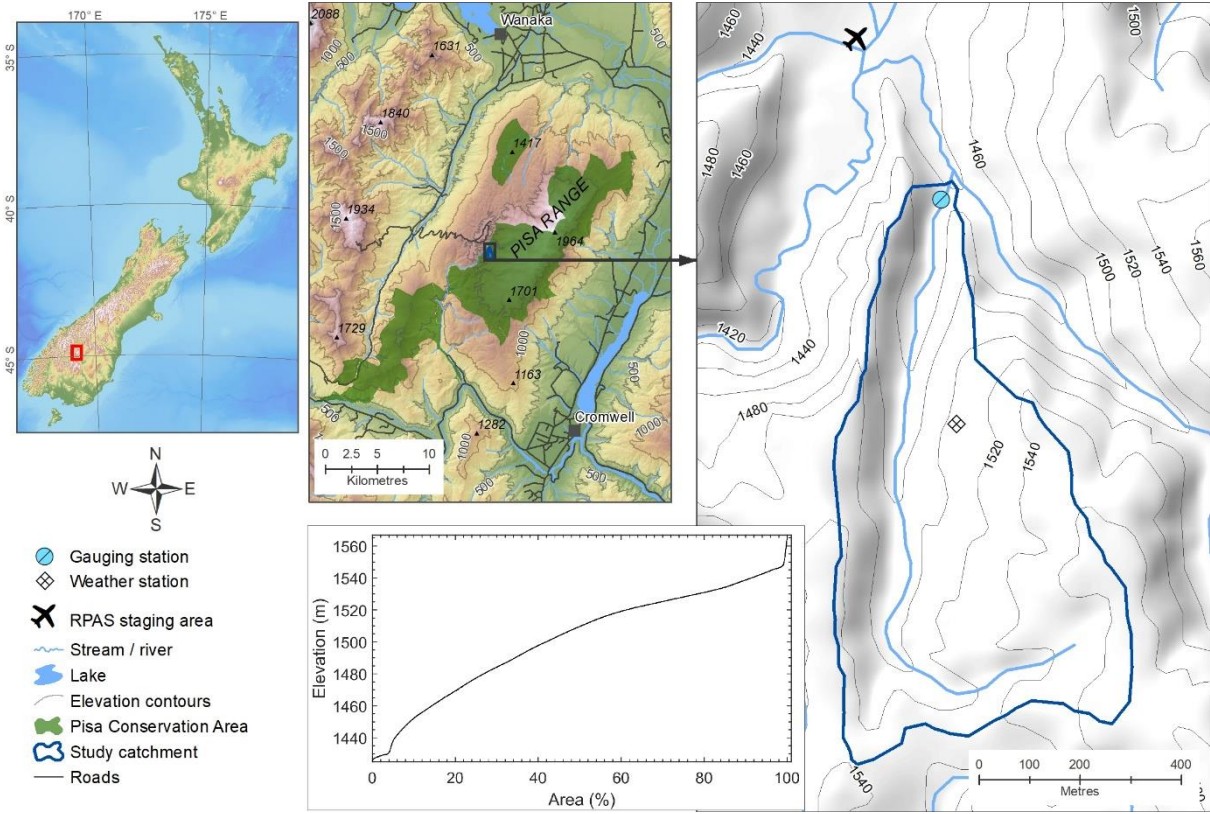

**Figure 1: Location and hypsometry of the study basin within the Pisa Range, New Zealand.**





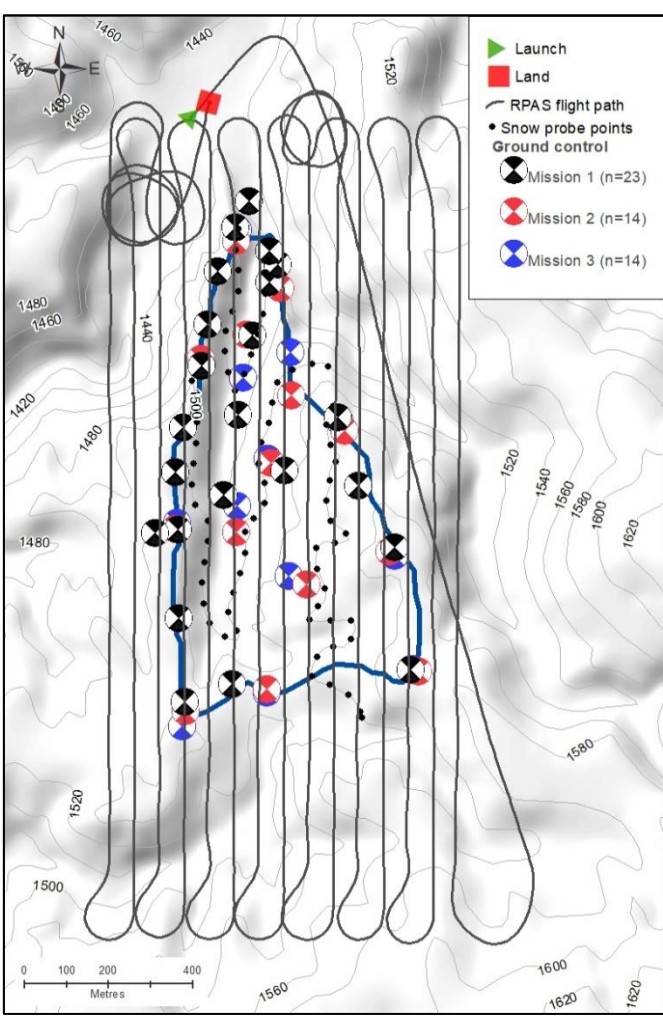

**Figure 2: Typical flight path for the mapping of the study basin using the Trimble UX5, GCP network established for each flight mission, and reference snow depth locations. Flight log is from the spring flight mission.**



**Figure 3: Processed ortho-mosaics for winter (A) and spring (B) flights, with corresponding maps of snow depth derived for winter (C) and spring (D).**





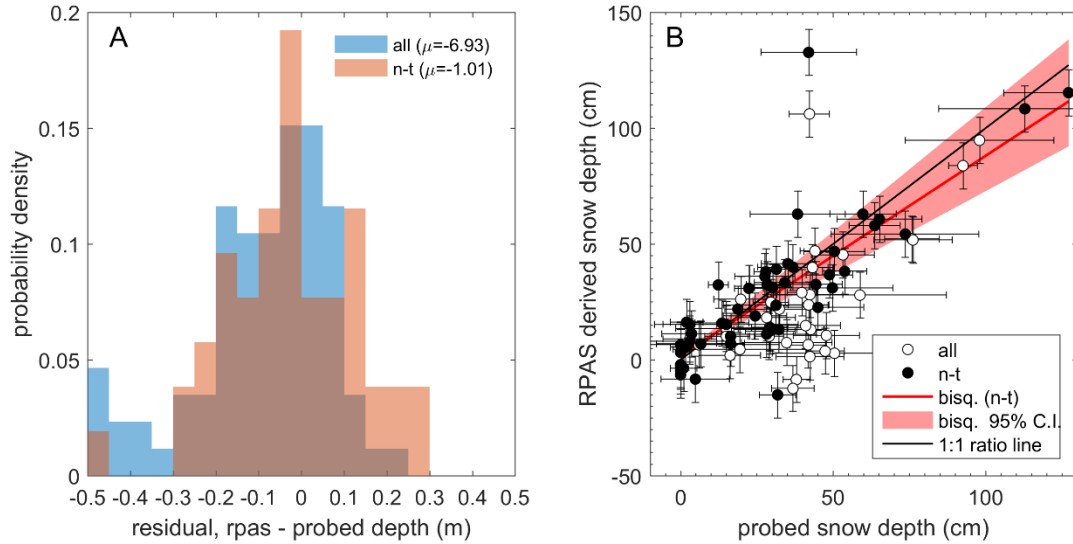

**Figure 4: Residuals between snow depths measured by RPAS photogrammetry and probing for all probe locations ("all", blue) and non-tussock probe locations ("n-t", red) (A), and bisquare weighted regression between snow depth derived from a 0.15 m RPAS grid and probed snow depths (B). Vertical error bars are determined from the error propagation associated with DSM differencing and have magnitude of ±0.094 m, while horizontal error bars are calculated from the standard deviation of probe measurements made at each reference sampling location.**





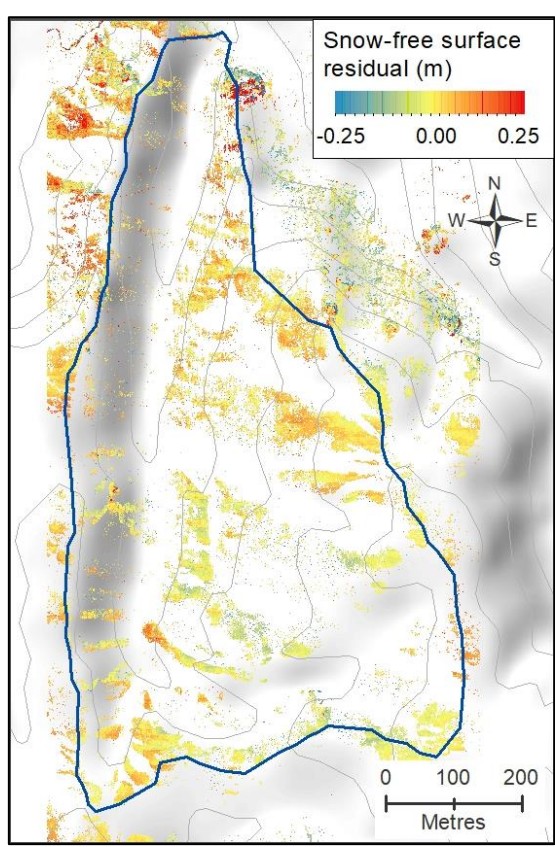

**Figure 5: Map of the vertical residual for snow-free areas for surface models derived from the autumn and spring flights.**



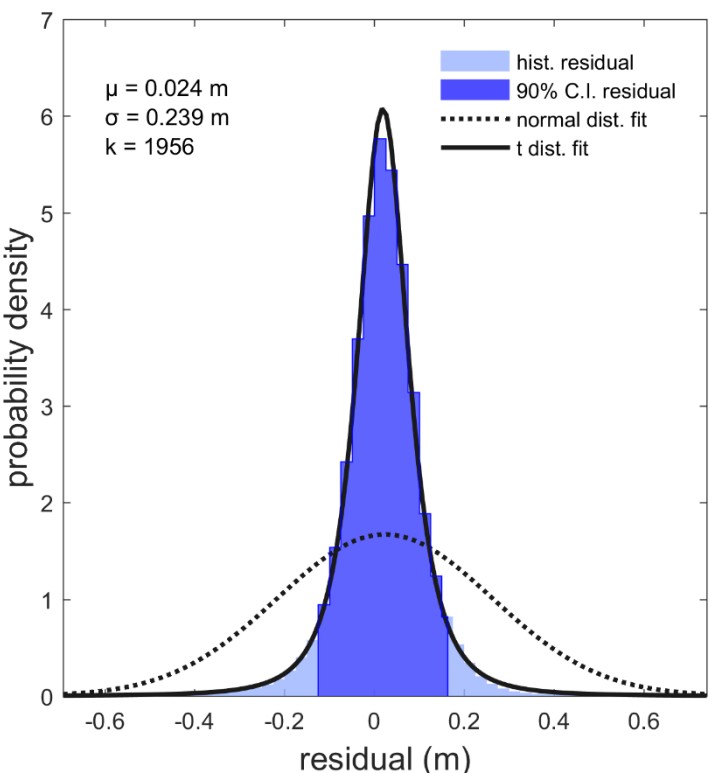

**Figure 6: Histogram of residuals for snow-free areas following differencing of autumn and spring DSMs, including fitted normal and t distributions.**





**Figure 7: Mean (μ), standard deviation (σ) and distribution kurtosis for the residual, in terms of discrete classes of slope (5° width), up to the 90th percentile of slope. Kurtosis is plotted on a log scale, and is accompanied by a standard error of 606. The slope histogram has been clipped to the 90[th] percentile.**





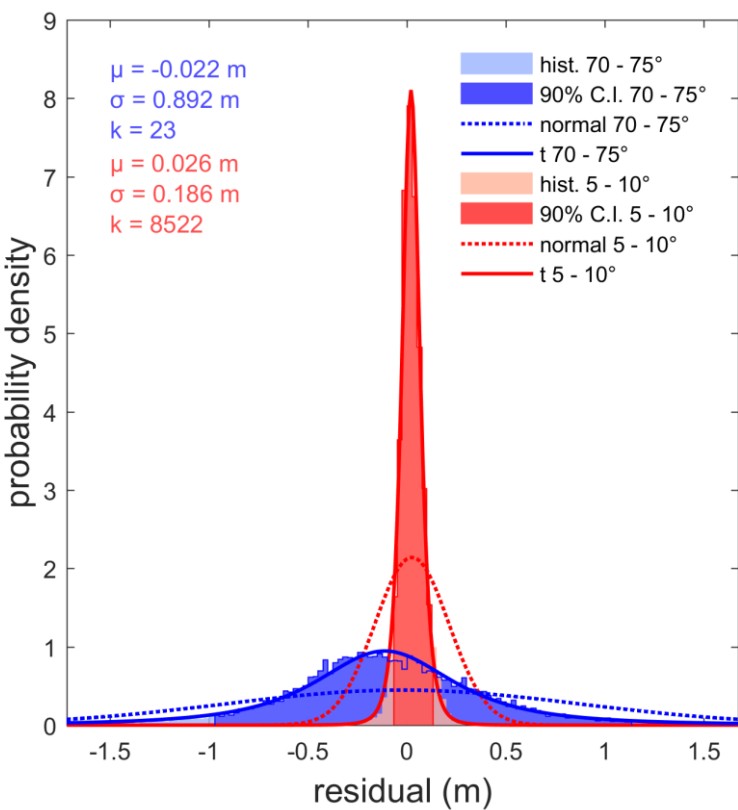

**Figure 8: Comparison of histograms and accompanying descriptive statistics for the residual between DSMs for slopes between 5 and 10° and slopes between 70 and 75°.**



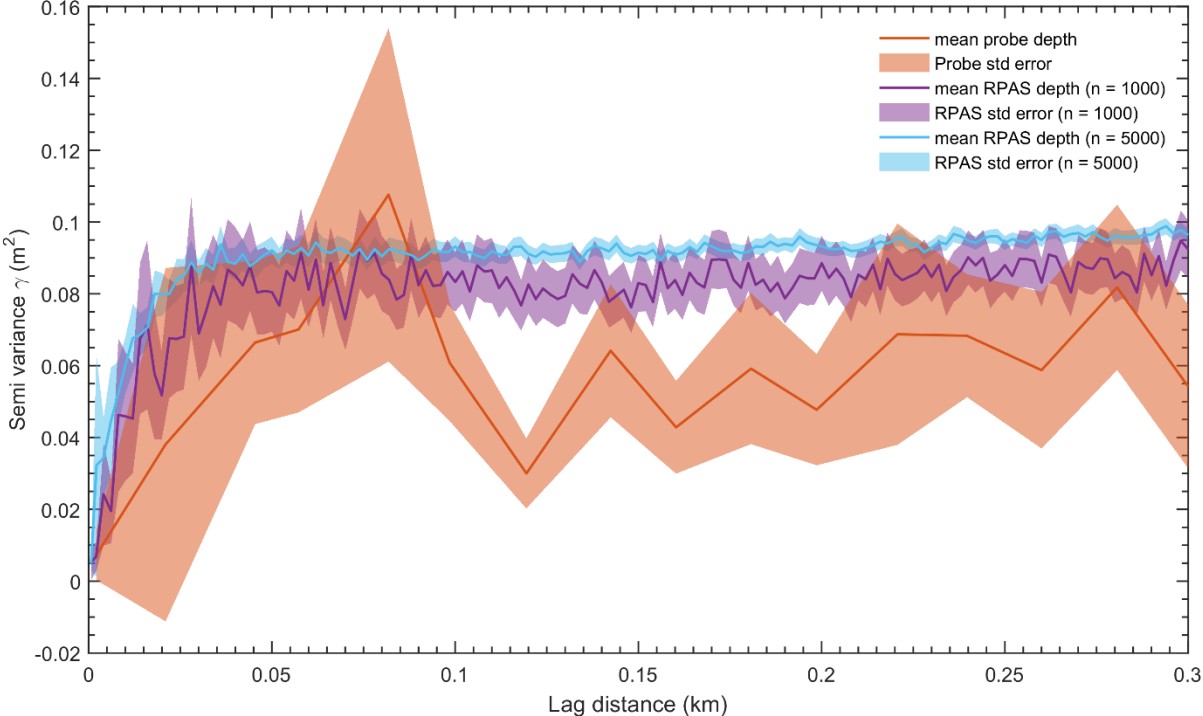

**Figure 9: Figure 9: Semi-variograms for snow depth, based on measurements provided by probing (86 samples), and two random samples drawn from RPAS-derived snow depth of 1000 and 5000 observations.**





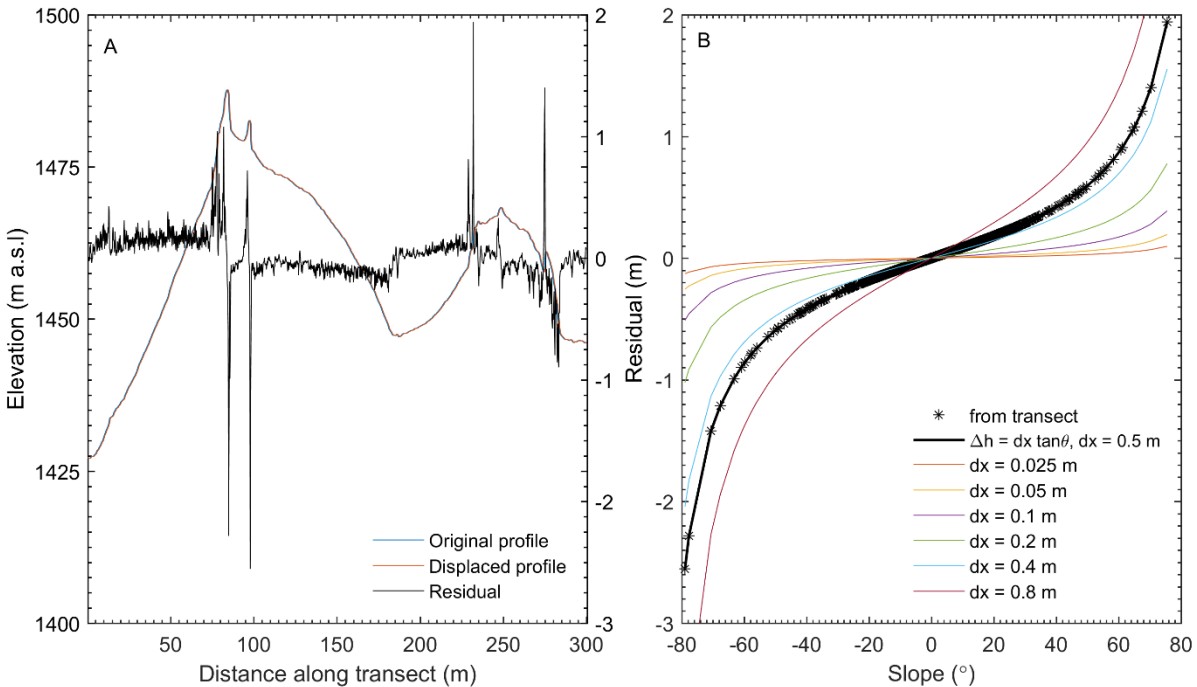

**Figure 10: The vertical residual between two elevation profiles, extracted from the same DSM, along a common transect, and offset horizontally by 0.5 m (A), and the resulting residuals plotted as a function of terrain surface slope, for the applied offset of 0.5 m, and a range of other offsets (B).**

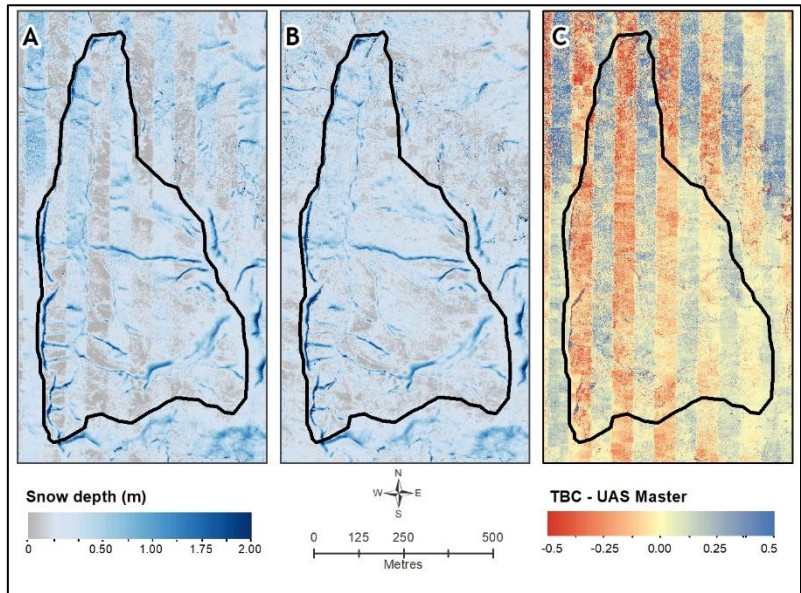

**Figure 11: Map of the systematic artefacts in dh (expected to represent snow depth), propagated when differencing DSMs resulting from aerial-triangulation in TBC v3.40 (A) compared with the dDSM from UAS Master (B). Vertical (north – south aligned) striping is highlighted in (C), the residual between dDSMs derived from TBC and UAS Master .**

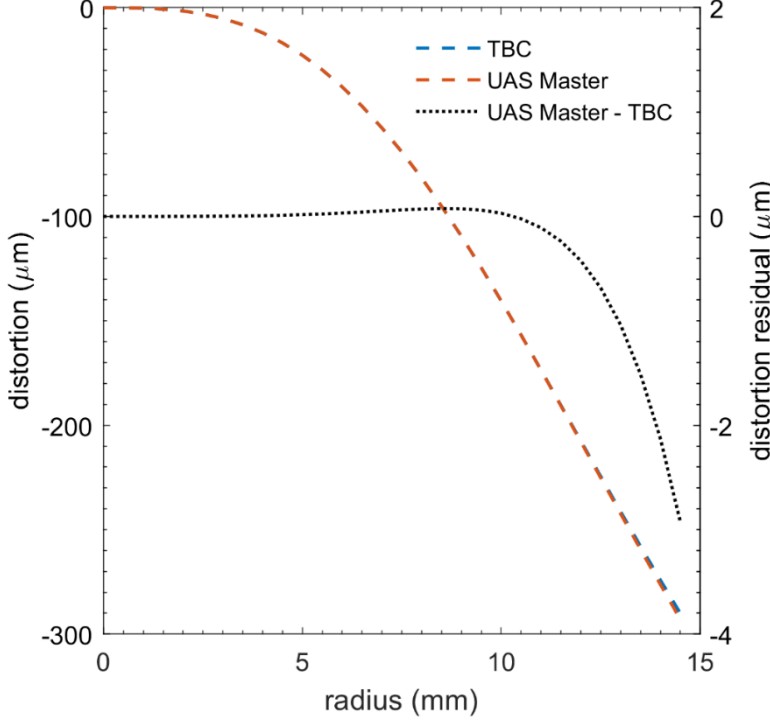

**Figure 12: Comparison of lens distortion characterised by individual triangulations of data from the same flight carried out in two different software packages, TBC and UAS Master. The residual was only apparent at radial distances >12 mm.**





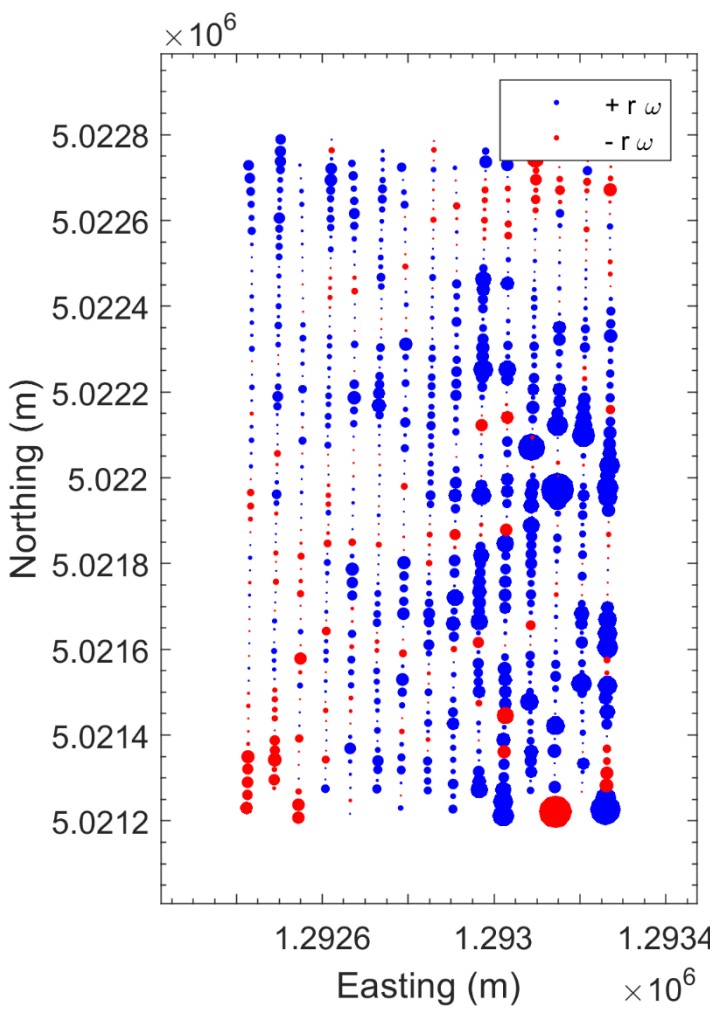

**Figure 13: Spatial distribution of residual in ω between TBC and UAS Master.**





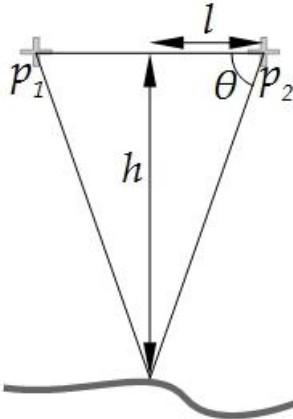

**Figure 14: Schematic of the relationship between h, l, and θ for a terrain point position resected from images centred at p1 and p2, when $\overline{r\theta}$ is small (e.g., 0.014° in this case).**

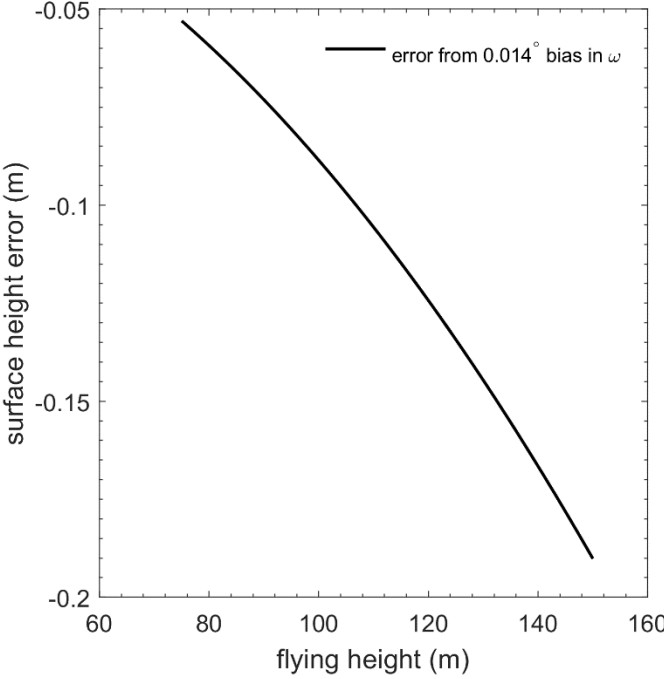

5    **Figure 15: Error in surface height propagated by a bias in ω, in relation to flying height.**