# Peer review of "Repeat mapping of snow depth across an alpine catchment with RPAS photogrammetry"

_The Cryosphere, 2018_

## Referee Comment (RC1) · Anonymous Referee #1 · 26 Jun 2018

**Initial paragraph or section evaluating the overall quality of the discussion paper ("general comments")**

This manuscript aims to present a methodology combining RPAS and photogrammetry to capture high-resolution snow depth estimates, to provide its accuracy and associated statistics. I believe the study itself has some strengths and interesting aspects; they cover a relatively large area catchment area using a fixed-wing UAS (compared to most studies using rotor based UAS), there is a considerable elevation change, it provides some distinct statistical investigations, and is done for 2 epochs.

However, in my opinion the authors have overlooked recent literature treating the same subject, and therefore have framed the study on a too general context, lacking some novelty and specific aims within the research of this methodology.  Furthermore, there are also some other scientific comments, which I will try to inquire further.

Therefore, I recommend major revisions to provide the opportunity to re-orient the focus and frame of their study, highlighting the strengths and further clarify on some more specific aspects of the method noted in literature.

Technically, I think the manuscript embodies a well-presented and carried-on study. Some structural changes and clarifications are recommended.

**Section addressing individual scientific questions/issues ("specific comments")**

TITLE AND OVERALL AIMS OF THE STUDY:

I think the title and aim are too general considering that there are already several studies investigating the same snow depth mapping method, looking at similar datasets (multiple DEMs + GCPs + snow probing validation). Furthermore, very similar validation methods have already been applied in other studies for multiple types of terrain/snow and providing almost the same results in terms of accuracy and resolution (although some for smaller areas). The manuscript here presented has referenced and discussed only four of them, but some additional studies on the method are missing:

*Avanzi, F.; Bianchi, A.; Cina, A.; De Michele, C.; Maschio, P.; Pagliari, D.; Passoni, D.; Pinto, L.; Piras, M.; Rossi, L. Centimetric Accuracy in Snow Depth Using Unmanned Aerial System Photogrammetry and a MultiStation.* Remote Sens. **2018**, *10*, *765.*

*Yves Bühler, Marc S. Adams, Andreas Stoffel & Ruedi Boesch (2017) Photogrammetric reconstruction of homogenous snow surfaces in alpine terrain applying near-infrared UAS imagery, International Journal of Remote Sensing, 38:8-10, 3135-3158, DOI: 10.1080/01431161.2016.1275060*

*Cimoli, E.; Marcer, M.; Vandecrux, B.; Bøggild, C.E.; Williams, G.; Simonsen, S.B. Application of Low-Cost UASs and Digital Photogrammetry for High-Resolution Snow Depth Mapping in the Arctic.* Remote Sens. **2017**, *9*, *1144.*

*Avanzi, F., Bianchi, A., Cina, A., De Michele, C., Maschio, P., Pagliari, D., Passoni, D., Pinto, L., Piras, M., and Rossi, L.: Measuring the snowpack depth with Unmanned Aerial System photogrammetry: comparison with manual probing and a 3D laser scanning over a sample plot, The Cryosphere Discuss., https://doi.org/10.5194/tc-2017-57, 2017.*

*Fernandes, R., Prevost, C., Canisius, F., Leblanc, S. G., Maloley, M., Oakes, S., Holman, K., and Knudby, A.: Monitoring snow depth change across a range of landscapes with ephemeral snow packs using Structure from Motion applied to lightweight unmanned aerial vehicle videos, The Cryosphere Discuss., https://doi.org/10.5194/tc-2018-82, in review, 2018.*

I believe it would be good to have a look at these, and better frame the motivation and title of the study to the suggested gaps in the research stated in literature. Snow depth mapping with RPAS as you present it is not so novel, and accuracy has already been assessed in multiple ways with similar (or higher) sample sizes. Nevertheless, this study can emphasize other really interesting aspects (e.g. larger area, fixed wing and focus on discussing other statistics as already partially done) but limitations needs to be discussed and literature accounted.

An example of a more specific title suggestion could be:

"Investigating snow depth retrieval of an alpine catchment area using photogrammetry and fixed-wing RPAS"

**1. INTRODUCTION:**

Paragraph starting at line 23: there might be too much information on satellites and SCA. I think snow depth mapping with RPAS is not directly comparable with satellite in terms of applications and uses, therefore I would personally not give too much weight on comparing both. In addition, I think you speak a lot about SCA in the introduction, but you do not mention its retrieval as part of your objectives and rarely speak about it again during your methods, results and so on. So maybe part of it is avoidable, I guess your focus is snow depth here and SCA is just intrinsically bounded to it.

**2. STUDY SITE:**

Here a lot has been said about the topography, but what about the snow?

It is important to add information about the daylight conditions of the surveys, and the snow textures/types encountered. It is recognized that these can affect SfM reconstructions on snow, and potentially the final DEM product. Maybe add to Table 1 and describe change in snow conditions over time.

In my opinion, an interesting aspect of your study area is the variable slope, and the different types of snow encountered over the epochs (if different), so highlighting these over time and space would be good (as you have already partially done).

Finally, It would be great to add to Figure 1 (or as you see fit) the orthophoto of the Autumn DEM, this is not present in the manuscript and its very important to better showcase your terrain DEM, vegetation areas and so on.

**3. DATA AND METHODS:**

**3.1.1 RPAS platform and payload**

What is the cost of the unit? You mention is low-cost, but compared to what? It does not sound like a very low cost system within the realm of drones (particularly if you consider DIY solutions), but it is low-cost compared to manned aerial systems. Please specify. Please use SI units, e.g. meters and not feet. Also, I does not seem you have specified what was your average flying altitude for each flight.

**3.1.2 RPAS flights**

Maybe more information like frame rate, software used for mission planning, how the missing was planned in such scenario etc. would be good.

**3.1.3 Ground control survey**

Same here, some information on the software used for processing GNSS data, and a reference (if available) would be good (e.g. the manual).

In addition, it is not clear from the text if you used some of the points for co-registration of the multiple DSMs (I understand you did not, why?). Particularly since you speak about co-registration later, in chapter 5.2.2. Please make this clear in the text and if yes, display these points on one of the figures.

**3.1.4 In situ snow depth measurements**

I would call this chapter "snow probing validation" or something else similar.

Also, please specify the GNSS accuracy of your RTK snow depth samples, and how are they representative of your claimed spatial resolution (0.15 m) if they are averaged over and arm distance of a meter or so.

**3. DATA PROCESSING**

**3.2.1 Photogrammetric processing**

Is all the formulation needed? Since all the calculations are carried on by a black-box software, maybe only a couple of references on the general photogrammetric principle should suffice.

In addition, many of the variables/parameters in the equations are not defined and need to be specifically declared in the text.

This section could also be coupled and properly merged with the following one, "software". (I assume they are black-box/off the shelf software, but little to none information is provided in this manuscript, particularly as you heavily discuss it in the discussion section).

**3.2.2 Software**

Since the software's employed are out of the commonly used by the UAS snow depth mapping community, maybe a small introduction on how they compare to Photoscan (or others) and some references on them would be beneficial. Particularly as they are heavily discussed further on.

An example: how do you locate GCPs in the DEM for this software, manually or automatically? From personal experience, this can be a considerable source of error and is worth mentioning.

The paragraphs following line 29, are somewhat confusing. Does the process of removing GCPs you mention refers only to the autumn mapping? Is it an analysis that was performed before you place the GCPs on the other epochs? I think I understood, but please clarify this on the text.

In addition, you do not mention which specific GCPs you removed. CPs RMSE values would probably change if you select a different combination of points, of the same sample size (e.g. 14), among your set. This is because overall total CP RMSE will be dependent not only on the number of GCPs you have used in your model, but also on their location, their accuracy in image identification and GPS precision. Is not clear if you tested this or not.

An interesting statistical approach would be to do a CPs validation using a bootstrap or similar method. By this I mean selecting random samples out of your GCPs and using them as CPs, multiple times. This would actually be an interesting approach to take that no study snow mapping study with RPAS has undertaken before. This for example be an interesting addition to your statistical investigation that has not been undertaken by previous studies.

**3.2.3 Deliverables**

I think this section would be better moved to the beginning of results, to outline all the final output retrieved.

**3.3 Quality and accuracy assessment**

Three methods for assessing the accuracy of the method are here employed; calculated error propagation, manual snow probing and snow free validation. I agree that combination of all of them helps assessing the overall accuracy of your results. However, each of them carry some limitations in this case, which require to be mentioned.

First, as RPAS photogrammetry provides very high spatial resolution measurements, I think the best way to obtain a strong accuracy estimate would be to compare it with other high spatial resolution estimates such as Lidar. Therefore, it is important to note that the validation applied in this study still lacks such estimate comparison.

Regarding validation tests for snow free areas, you need to mention that this validation is mostly representative of snow free areas, and not much of the snow-covered areas. Particularly since you have not mentioned anything about the snow type and about the photogrammetric reconstruction performance.  Snow free areas can help assessing if the DSMs used for subtraction are shifted horizontally, and partially support you DSM reconstruction, but they cannot be used as a rigorous estimate of accuracy of snow depth mapping. I think is just important to mention this caveat.

Finally, in line 3 of page 9, you highlight the calculation of error propagation as the most rigorous validation among them and a bonus of this study. In my opinion, is the actual snow probing that validates better and independently your dDSM, particularly because for your calculation you have only a sample of 6 CPs each as I understand, whereas you have 86 snow depth probing samples.

**3.3.1 Uncertainty associated with RPAS-derived snow depth**

Please mention why did you assume that planimetric precision of each constituent DSM is negligible. In my experience, if you subtract two DSMs and there is just a slight misalignment, errors in the final dDSM (and bulk snow pack volume) can be considerable. That is why other studies have shown that co-georeferencing the DSMs using common GCPs can improve accuracy of the method.

**3.3.3 Repeatability of photogrammetric modelling**

Please describe a little bit more what you have done in your classification algorithm.

What do you mean by $dDSM_3$? I can't find this DSM identifier anywhere else in the text/tables.

**4. RESULTS**

**4.2.2 Assessment against reference probe data**

It would be great if you can also display on map the difference between snow probed and RPAS estimated snow depth for each of the 86 sample points. That would give a spatial overview of the error, or across the slope for example. It could also open up for discussion of accuracy across slope and epochs or snow types.

**4.2.3 Comparison of DSMs from independent RPAS flights**

I would suggest merging Figure 6 and Figure 5 somehow. I think the manuscript has already too many Figures that can be reduced.

**5. DISCUSSION**

**5.1 Performance of RPAS photogrammetry for resolving snow depth**

I believe here you highlight well the main strength of your study; that is a relatively larger area and mapped at high resolution. This is interesting and perhaps your study should be more centered on this. However, you should highlight what are some of the limitations in your validations here.

Line 3, page 14: "Achievement of uncertainties <0.13 m": Please mention to which statistics you refer. I guess the propagation of error?.

**5.2.1 Vegetation**

This has been highlighted by every study on snow depth mapping from RPAS, so should not be much of a novel finding discussion topic. I would however mention how it affected your dataset (as you already do), how it affected other studies and stress that solutions are really needed to solve this important caveat of the methodology.

It would be interesting to know instead how did error varies with slope or with snow type spatially across your catchment area and across epochs (if any difference is noticeable).

**5.2.2 Geo-location and co-registration**

It would be good if the authors stress more clearly/directly/shortly what is the importance of all this and its relevance in a RPAS snow-mapping context. What I understand here is that the authors first want to point out that vertical dDSM uncertainties and errors increase when planimetric and horizontal geo-location errors are present, and particularly on steep slopes or presence of rock outcrops (not surprisingly). Then the aim to is to justify the found statistics in this study, particularly for the slope and this is interesting, but requires a more clearer explanation of the benefits and consequences.

The main issue here is that is claimed that utilizing independent aero triangulation is better that co-georeferencing the DSMs. I personally disagree with this statemet, co –georeferencing saves considerable time when snow free areas are available. And if not available, it's always easier to leave artificial features like high poles (seen all over the year), which require only 1 measurement for the entire seasonal snow cycle over multiple years. Many studies have reported that co-georeferencing their terrain and snow DSMs using common GCPs considerably improved the snow depth maps accuracy performance (because of all the problems you mentioned). They mitigate considerably all the issues of planimetric misalignment.

In addition, I would see this discussion to be more focused on snow itself, and its changes in aspect, and elevation in accordance with underlying terrain and how this will affect statistics of other people employing the method, under different snow landscapes. This rather than focusing on a general DSM context.

Finally, you mention that high quality GCPs are important, but you don't discuss the future of systems with on-board RTK which will probably substitute GCPs in the near future and can account more directly and precisely the errors you mention in roll, pitch and yaw by integrating IMU components.

**5.3 Pitfalls and limitation of RPAS photogrammetry**

This large sub-chapter and more than three figures are dedicated to comparing the performance of two different black-box photogrammetric software's. I guess this is not part of your initial aims, and

while is partially interesting for some communities, I think your goal is to outline a snow depth mapping method and not the performance of the particular software's. Therefore, I would personally reduce considerably this section. Particularly since inferences are made based on two different black-box software's and you don't really comment/outline/reference on the particular workflow of these products.

Another issue is that here you generalize as "limitations and pitfalls of RPAS photogrammetry" problems that were encountered within the particular software employed. Other studies have not reported these discussed issues with other software that I am aware (or they could not be verified?). Therefore, is difficult generalize such a discussion for snow depth mapping with RPAS.

5.4 Spatial and temporal trends in snow cover

This section much better highlights the strengths of your study and more should be reflected in the introduction and aims. The same can be said about your conclusion. Nevertheless, more could be said about the high repeatability/change detection potential of your methodology, since your covered 2 distinct epochs.

**Purely technical corrections at the very end ("technical corrections": typing errors, etc.).**

Please make units consistent thorough the manuscript. Either cm or m (e.g. line 4 of page 6).

In all tables and figures, please include full names and abbreviations. Often, only abbreviations are shown and the reader is forced to look for them in the text.

Line 4 page 11: There is clearly a typographical error.

In figure 2 please add "points" to Ground Control…

---

## Referee Comment (RC2) · Anonymous Referee #2 · 1 Jul 2018

This study presents results for a small watershed in New Zealand where repeat unmanned aircraft flights were used to map surface elevations using photogrammetric methods, and then snow depth via digital surface model differencing. There was one snow free flight, and two snow on flights, one winter and one spring. Although the snow depth results are presented, the main focus of the paper is more technically focused on methods, uncertainty, and validation.

The use of unmanned aerial systems in earth science is growing in popularity for good reason; the units are small, relatively inexpensive, easy to deploy, and the software to carry out structure from motion photogrammetry is becoming more accessible and user friendly. This study is a relevant and useful contribution to the growing body of literature using UAS to map snow depth and cryospheric processes at high resolution,

fits within the scope of The Cryosphere, and should be accepted for publication after revisions. Following are broad recommendations that would improve the manuscript, namely in terms of readability and accessibility by a broader audience, particularly one that may not be familiar with mapping surface elevations/snow depth with UAS.

- The acronym RPAS was new to me, likely a regional difference in terminology that I am unfamiliar with. In terms of search-ability I would suggest the switch to UAV or UAS (which is already used in the paper- so that would simplify things), or at minimum, mention the different terms use for unmanned aerial systems in the introduction and justify the use of RPAS rather than UAS.

- The manuscript reads as if the authors assume the reader has some understanding of photogrammetry, which is not necessarily a safe assumption. Something as simple as 'overlapping pictures are used to reconstruct a continuous 3 dimensional surface' very early on in the introduction would be helpful to provide context to the reader, and also making sure important terms are defined (like tie point). Also aerotriangulation is simply the georeferencing method by which ground control values are assigned to points, this could be defined once and then the term georeferencing could be used afterwards, which is a more accessible term. This paper dives into the technical very quick, but shouldn't forget to cover the basics, as well, since this is still a relatively new method for mapping snow depth.

- This paper does a great job of covering uncertainty, but I think it is interesting and important to recognize the practical limitations of this method early on in the paper. It currently cannot scale up beyond small watersheds for practical reasons, namely flight times and flight restrictions, which vary widely from country to country. Also setting out ground control points can be just as time consuming and limiting as carrying out snow surveys, which is why the authors themselves wanted to reduce the numbers of GCPs used per flight. Also vegetation is a critical issue in watersheds that have thick brush, or trees for that matter, so it is only useful and accurate in alpine watersheds. Discussing how these issues might be be overcome in the future to make this method operationally

useful would be very interesting (i.e. that use of RTK on the UAS). As it stands snow in medium to large scale, and/or vegetated, watersheds can only be mapped with lidar, and while it is notably missing in the paper, repeat high resolution lidar flights for snow depth and SWE are being done in the Western US by the Airborne Snow Observatory at operationally relevant scales (https://doi.org/10.1016/j.rse.2016.06.018).

-In the introduction the authors emphasize how valuable this method could be for understanding spatial variability in snow depth at high resolution, but then spend very little time actually presenting snow depth results for the two snow-on flights. I do think the uncertainty discussion is important and relevant, but so is the snow depth results, and more time should be spent on them. Also, snow water equivalent is only mentioned briefly at the end, this should be an entire results in the section and the measurement of densities should be covered in the methods. An estimate of SWE for the two flights would be really interesting. (Minor note, on pg 9, line 29 the authors say the nominal accuracy for snow probes is +/- 1 cm, if this is from the literature it should have a citation, because I understand it to be much larger due mostly to user error, which they themselves recognize, in detail, later.)

-It is quite obvious that one of the authors has a thorough understanding of statistics. It gets tedious, and in these sections/figures most readers will just skip over. I would suggest for each relevant result adding 1 plain language summary before diving into the details to improve readability. 'Uncertainty is larger for more rapid changes in topography'.

-It is not clear to me why the authors spend so much space in terms of text and figures on georeferencing errors with older software when it could be covered in a few sentences, and more time could spent on more relevant results (i.e. the gist of this is that the old software had large errors, the new software performs better, so the old software should be avoided). This would also reduce the number of figures (there are so many).

- General editing comments: Writing structure and grammar need some attention, as

they were notable enough to distract from the science being presented. The first paragraph of the Intro needs to be rewritten to read more consistently and should introduce the context and motivation for this study specifically. All paragraphs should be at least three sentences in length. There are many run on sentences that made reading and interpreting intent challenging. Watch out for the use of colloquial terms in a scientific context ('hamper' or 'impair' for something that is a challenge or difficult, the use of the word 'see' or 'saw' for things that don't have eyes). A small but related note, I associate the term epoch with geologic time scales (a division of time that is a subdivision of a period and is itself subdivided into ages, corresponding to a series in chronostratigraphy), I suggest not using this term and in most places through out the text it is unnecessary. For overall readability of the technical sections it might be useful to think about what content contributes to the overall purpose of the study given the audience (like equations 1-5, I don't find these critical to include, interested readers could be provided with a reference to follow up with). It maybe useful to have someone that is a physical scientist, but not involved in the study, read through the paper and give feedback.

---

## Author Comment (AC1) · 7 Sep 2018

**Mapping snow depth at very high resolution with RPAS photogrammetry**

Todd A. N. Redpath, Pascal Sirguey, Nicolas J. Cullen

**Response to reviewer comments**

We thank the referees for their careful review and helpful feedback, from which we have improved the manuscript.

*Referee comments in italic font*

**Response** In normal font.

**1. Response to RC1 comments**

*Initial paragraph or section evaluating the overall quality of the discussion paper ("general comments")*

*This manuscript aims to present a methodology combining RPAS and photogrammetry to capture high-resolution snow depth estimates, to provide its accuracy and associated statistics. I believe the study itself has some strengths and interesting aspects; they cover a relatively large area catchment area using a fixed-wing UAS (compared to most studies using rotor based UAS), there is a considerable elevation change, it provides some distinct statistical investigations, and is done for 2 epochs.*

*However, in my opinion the authors have overlooked recent literature treating the same subject, and therefore have framed the study on a too general context, lacking some novelty and specific aims within the research of this methodology. Furthermore, there are also some other scientific comments, which I will try to inquire further.*

*Therefore, I recommend major revisions to provide the opportunity to re-orient the focus and frame of their study, highlighting the strengths and further clarify on some more specific aspects of the method noted in literature.*

*Technically, I think the manuscript embodies a well-presented and carried-on study. Some structural changes and clarifications are recommended.*

**Response** We thank the reviewer for their comments and thorough review.

*Section addressing individual scientific questions/issues ("specific comments")*

*TITLE AND OVERALL AIMS OF THE STUDY:*

*I think the title and aim are too general considering that there are already several studies investigating the same snow depth mapping method, looking at similar datasets (multiple DEMs + GCPs + snow probing validation). Furthermore, very similar validation methods have already been applied in other studies for multiple types of terrain/snow and providing almost the same results in terms of accuracy and resolution (although some for smaller areas). The manuscript here presented has referenced and discussed only four of them, but some additional studies on the method are missing:*

*Avanzi, F.; Bianchi, A.; Cina, A.; De Michele, C.; Maschio, P.; Pagliari, D.; Passoni, D.; Pinto, L.; Piras, M.; Rossi, L. Centimetric Accuracy in Snow Depth Using Unmanned Aerial System Photogrammetry and a MultiStation. Remote Sens. 2018, 10, 765.*

*Yves Bühler, Marc S. Adams, Andreas Stoffel & Ruedi Boesch (2017) Photogrammetric reconstruction of homogenous snow surfaces in alpine terrain applying near-infrared UAS imagery, International Journal of Remote Sensing, 38:8-10, 3135-3158, DOI: 10.1080/01431161.2016.1275060*

*Cimoli, E.; Marcer, M.; Vandecrux, B.; Bøggild, C.E.; Williams, G.; Simonsen, S.B. Application of Low-Cost UASs and Digital Photogrammetry for High-Resolution Snow Depth Mapping in the Arctic.* Remote Sens. *2017*, 9, *1144.*

*Avanzi, F., Bianchi, A., Cina, A., De Michele, C., Maschio, P., Pagliari, D., Passoni, D., Pinto, L., Piras, M., and Rossi, L.: Measuring the snowpack depth with Unmanned Aerial System photogrammetry: comparison with manual probing and a 3D laser scanning over a sample plot, The Cryosphere Discuss., https://doi.org/10.5194/tc2017-57, 2017.*

*Fernandes, R., Prevost, C., Canisius, F., Leblanc, S. G., Maloley, M., Oakes, S., Holman, K., and Knudby, A.:*
*Monitoring snow depth change across a range of landscapes with ephemeral snow packs using Structure from Motion applied to lightweight unmanned aerial vehicle videos, The Cryosphere Discuss., https://doi.org/10.5194/tc-2018-82, in review, 2018.*

*I believe it would be good to have a look at these, and better frame the motivation and title of the study to the suggested gaps in the research stated in literature. Snow depth mapping with RPAS as you present it is not so novel, and accuracy has already been assessed in multiple ways with similar (or higher) sample sizes. Nevertheless, this study can emphasize other really interesting aspects (e.g. larger area, fixed wing and focus on discussing other statistics as already partially done) but limitations needs to be discussed and literature accounted.*

**Response** We thank the reviewer for this constructive and well-informed suggestion. We agree that the point of difference brought by our study, namely the use of fixed-wing over an entire alpine catchment, could be strengthened. We have substantially reworked and widened the scope of the introduction to include some of the recent work suggested by the reviewer, and have emphasised our focus on mapping snow depth over a larger area and substantially more complex terrain compared to what has been reported in previous studies. We identify the additional set of challenges associated with the larger area and increased relief to justify the need for methodological assessment. From the citations proposed by the reviewer, we note that Avanzi et al. (2018) is arguably identical to Avanzi et al. (2017) and thus cited only the version accepted in Remote Sensing rather that the discussion paper in The Cryosphere Discussion.

*An example of a more specific title suggestion could be:*

*"Investigating snow depth retrieval of an alpine catchment area using photogrammetry and fixedwing RPAS"*

**Response** We agree the focus of the paper could be better conveyed in the title and appreciate the suggestion of the reviewer. We amended the title to "Repeat mapping of snow depth across an alpine catchment with RPAS photogrammetry"

**1. *INTRODUCTION:**

*Paragraph starting at line 23: there might be too much information on satellites and SCA. I think snow depth mapping with RPAS is not directly comparable with satellite in terms of applications and uses, therefore I would personally not give too much weight on comparing both. In addition, I think you speak a lot about SCA in the introduction, but you do not mention its retrieval as part of your objectives and rarely speak about it again during your methods, results and so on. So maybe part of it is avoidable, I guess your focus is snow depth here and SCA is just intrinsically bounded to it.*

**Author response** This section of the introduction has been substantially reduced. We maintain some reduced context around remote sensing of SCA and other snow metrics as there is motivation in assessing how insight from high resolution data such as RPAS photogrammetry may improve inferences made from coarser space-borne products.

2. *STUDY SITE:*

*Here a lot has been said about the topography, but what about the snow?*

**Response** At the time this study was planned, very little data was available regarding snow within the study basin and surrounding areas, with the exception of the work of Sims and Orwin (2011) – this is a widespread problem in New Zealand. As noted, visual inspection of available satellite imagery was undertaken to assess the viability of the field site from a snow cover point of view. Ultimately, the topography of this site also makes it of interest for the study of snow in New Zealand, with the morphology being somewhat different from the Southern Alps, but similar to many of the eastern high ranges of Otago, which are under-represented in snow observation and study.

*It is important to add information about the daylight conditions of the surveys, and the snow textures/types encountered. It is recognized that these can affect SfM reconstructions on snow, and potentially the final DEM product. Maybe add to Table 1 and describe change in snow conditions over time.*

**Response** Additional information has been added to Table 1 and Section 3.1.2 to further describe snow and sky/lighting conditions at the time of each flight. We agree that these are of interest in terms of the application of the methodology. We shall however stress that potential limitations are related more to the hardware used to capture imagery (in particular the dynamic range of the camera being used), rather than the photogrammetry modelling itself. We clarified this by reporting in Section 3.1.2 about the suitable contrast achieved by the camera.

*In my opinion, an interesting aspect of your study area is the variable slope, and the different types of snow encountered over the epochs (if different), so highlighting these over time and space would be good (as you have already partially done).*

*Finally, It would be great to add to Figure 1 (or as you see fit) the orthophoto of the Autumn DEM, this is not present in the manuscript and its very important to better showcase your terrain DEM, vegetation areas and so on.*

**Response** wW agree, and have added the orthophoto and hillshaded DEM from the Autumn flight to an additional panel in Figure 3.

*3. DATA AND METHODS:*

*3.1.1 RPAS platform and payload*

*What is the cost of the unit? You mention is low-cost, but compared to what? It does not sound like a very low cost system within the realm of drones (particularly if you consider DIY solutions), but it is low-cost compared to manned aerial systems. Please specify. Please use SI units, e.g. meters and not feet. Also, I does not seem you have specified what was your average flying altitude for each flight.*

**Response** Our reference to low cost in the introduction is not specific to the unit used in this study, but to the use of RPAS in general, particularly relative to manned platforms. We have clarified this in the introduction. We prefer to retain units of feet when referring to aircraft heights, as this is established practice by regulatory bodies and within the aviation sector, but have included conversions to metres. We have added the average flying altitude (which was the same for the three flights) to section 3.1.2.

**3.1.2 RPAS flights**

*Maybe more information like frame rate, software used for mission planning, how the missing was planned in such scenario etc. would be good.*

**Response** Agreed. we have specified the software used for flight mission planning (Trimble Aerial Imaging). The system we used does not rely on a fixed frame rate but rather on predetermined locations of exposure stations to achieve the desired forward overlap. GNSS navigation and infrared trigger on-board ensures images are captured at the appropriate location. We clarified this in section 3.1.2.

**3.1.3 Ground control survey**

*Same here, some information on the software used for processing GNSS data, and a reference (if available) would be good (e.g. the manual).*

**Response** The name of the software used for processing GNSS data (Trimble Business Center) has been added.

*In addition, it is not clear from the text if you used some of the points for co-registration of the multiple DSMs (I understand you did not, why?). Particularly since you speak about co-registration later, in chapter 5.2.2. Please make this clear in the text and if yes, display these points on one of the figures.*

**Author response** No control points were used as co-registration points. Although the location of GCPs between flights were similar, they were not identical. Variable snow cover obscuring surface features between different flights precluded co-registration of the DSMs. This has been clarified in the text.

**3.1.4 In situ snow depth measurements**

*I would call this chapter "snow probing validation" or something else similar.*

**Response** We have amended the heading to *Reference snow depth measurements*

*Also, please specify the GNSS accuracy of your RTK snow depth samples, and how are they representative of your claimed spatial resolution (0.15 m) if they are averaged over and arm distance of a meter or so.*

**Response** The probe locations were surveyed according to the same protocol as GCPs, and achieved the same level of accuracy; the text has been updated to reflect this. We have added a comment in section 4.2.2. to note the influence of spatial uncertainty on the comparison between RPAS and probed snow depths.

**3. DATA PROCESSING**

**3.2.1 Photogrammetric processing**

*Is all the formulation needed? Since all the calculations are carried on by a black-box software, maybe only a couple of references on the general photogrammetric principle should suffice.*

**Response** We believe that the main formulation of the photogrammetric model is desirable here as it provides context to discuss the sensitivity of the method to errors in the estimate of interior and/or exterior orientation parameters, which is addressed later in the paper. We also think that given the recent proliferation of RPAS applications, and in particular when relying on black/grey-box software solutions, it is important to stress the mathematical fundamentals of photogrammetric modelling. This is critical to understand potential pitfalls in the technique, yet may be unfamiliar to many readers. We have amended the text to emphasise the relevance of this.

*In addition, many of the variables/parameters in the equations are not defined and need to be specifically declared in the text.*

**Response** Agreed. We have defined the *K* and *T* terms which had been erroneously omitted previously. We also moved to this paragraph the definition of ($X_0$, $Y_0$, $Z_0$) which was provided latter in the manuscript.

*This section could also be coupled and properly merged with the following one, "software". (I assume they are black-box/off the shelf software, but little to none information is provided in this manuscript, particularly as you heavily discuss it in the discussion section).*

**Response** We believe that this section is best kept separate from the following, as it introduces the general principles of photogrammetry, while the following section relates specifically to the software used and workflow implemented.

*3.2.2 Software*

*Since the software's employed are out of the commonly used by the UAS snow depth mapping community, maybe a small introduction on how they compare to Photoscan (or others) and some references on them would be beneficial. Particularly as they are heavily discussed further on.*

**Response** We have added some more detail regarding the software here, as well as references to documentation. Note we have re-named this section "Software and workflow".

*An example: how do you locate GCPs in the DEM for this software, manually or automatically? From personal experience, this can be a considerable source of error and is worth mentioning.*

**Response** We agree that the process to collect GCPs is important to clarify. In our case the collection of signalled markers is manual and we have added relevant detail in the text.

*The paragraphs following line 29, are somewhat confusing. Does the process of removing GCPs you mention refers only to the autumn mapping? Is it an analysis that was performed before you place the GCPs on the other epochs? I think I understood, but please clarify this on the text.*

**Response** The text has been edited to clarify that this test was applied only to the autumn epoch.

*In addition, you do not mention which specific GCPs you removed. CPs RMSE values would probably change if you select a different combination of points, of the same sample size (e.g. 14), among your*

*set. This is because overall total CP RMSE will be dependent not only on the number of GCPs you have used in your model, but also on their location, their accuracy in image identification and GPS precision. Is not clear if you tested this or not.*

*An interesting statistical approach would be to do a CPs validation using a bootstrap or similar method. By this I mean selecting random samples out of your GCPs and using them as CPs, multiple times. This would actually be an interesting approach to take that no study snow mapping study with RPAS has undertaken before. This for example be an interesting addition to your statistical investigation that has not been undertaken by previous studies.*

**Response** We agree that the relative arrangement of GCPs and CPs is important to the quality of the triangulation, and have added this information to Figure 2. The objective was always to ensure that the perimeter of the basin was constrained by GCPs. We also agree with the reviewer on the merit of the proposed method, and as a matter of fact are used to leave-one-out cross-validation protocols to leverage efficiently limited GCP networks in photogrammetric projects. In this context however, it is important to note that the sheer number of images and TP involves a significant processing time for each adjustment which makes numerous repeats impractical. During early trials to test the robustness of our photogrammetric modelling on the autumn flight, we did in fact carry out a leave-one-out cross-validation when evaluating the performance of the TBC software, and found consistent AT results. However, as explained in the manuscript, despite such assessment the surfaces produced subsequently by TBC remained compromised, with errors not captured by the AT only revealed when differencing surfaces subsequently. This stresses the limitation of relying solely on GCP/CP residual reports as this may not capture all errors of the photogrammetric modelling. When reconsidering processing with a more capable software solution (UASMaster), we decided to repeat only a limited number of scenarios which we believed captured well the information needed to inform the GCP network design for winter flight, with the cross-validation protocol finally adding marginally to this assessment. In this case, the overarching objective remained to inform the minimum number of GCPs that might be needed to provide a robust triangulation in the context of this specific field site and the complications associated with the terrain and winter operations. The fact that we have differing spatial arrangements of GCPs and CPs for each of the flights, with comparable triangulation residuals despite independent networks, goes some way to illustrating this, and demonstrates that consistent results can be achieved in lieu of permanent, common, control marks in the field. Although we appreciate the well-informed comment of the reviewer, we believe our protocols remain sufficient to document robustness of the modelling and decided not to make changes to the manuscript to evaluate the suggestion of the reviewer.

*3.2.3 Deliverables*

*I think this section would be better moved to the beginning of results, to outline all the final output retrieved.*

**Response** In terms of deriving snow depth, these products are really an intermediary step. We considered shifting this section to the results, but that would lead to an inconsistency with section 3.2.4. Instead, we re-named this section *Intermediate deliverables*.

*3.3 Quality and accuracy assessment*

*Three methods for assessing the accuracy of the method are here employed; calculated error propagation, manual snow probing and snow free validation. I agree that combination of all of them*

*helps assessing the overall accuracy of your results. However, each of them carry some limitations in this case, which require to be mentioned.*

*First, as RPAS photogrammetry provides very high spatial resolution measurements, I think the best way to obtain a strong accuracy estimate would be to compare it with other high spatial resolution estimates such as Lidar. Therefore, it is important to note that the validation applied in this study still lacks such estimate comparison.*

**Response** We agree that Lidar would provide a useful reference measurement, however such data are not available for the study area, which we have noted in the text.

*Regarding validation tests for snow free areas, you need to mention that this validation is mostly representative of snow free areas, and not much of the snow-covered areas. Particularly since you have not mentioned anything about the snow type and about the photogrammetric reconstruction performance. Snow free areas can help assessing if the DSMs used for subtraction are shifted horizontally, and partially support you DSM reconstruction, but they cannot be used as a rigorous estimate of accuracy of snow depth mapping. I think is just important to mention this caveat.*

**Response** We agree that the validation is mostly representative of snow-free areas. We are confident, however, that these results are transferable to snow covered areas given that good contrast was obtained over snow surfaces where dense photogrammetric restitution was not impaired. We don't expect (and haven't observed) any noticeable degradation in photogrammetric performance over snow covered areas. DSMs are free of widespread noise, and spurious surface elevations are uncommon. We attribute this mostly to the characteristics of the camera used (Sony NEX 5R), which has an APS-C sized sensor (considerably larger than the cameras deployed in many other RPAS platforms used for mapping). With an effective resolution of 16 M pixels, the NEX 5R individual photodetectors are >3 times larger than those found on the Canon ELPH 110 (commonly carried by the Sensefly eBee), and the cameras fitted to DJI Phantom systems (1/2.3" 16.1 M pixel), and about 0.8 microns larger than that of the Sony NEX 7. A high sensitivity to light and careful selection of exposure settings before flight ensures contrast and good sharpness across images are maintained, with high dynamic range and minimal occurrence of under or over exposure. Indeed, our experience has been that where scenes are mixed between snow and snow-free areas, the snow-free areas tend to be relatively under-exposed relative to the snow, which is well resolved. For us, this is preferred, and flights over snow are always operated with a high shutter speed accordingly. We have included sample image frames here to demonstrate the surface contrast achieved under varying snow conditions. Note also that we flew with a NIR modified NEX 5R for the winter flight, but this did not provide any substantial improvement over the RGB for the triangulation. Comparative study between modelling restitution of NIR vs RGB acquisitions are however not in the scope of this paper. In view of this, we therefore maintain that testing repeatability on snow-free areas informs equally on the robustness, consistency and performance of modelling over snow when such contrast is obtained.

[Figure]

*Figure 1: Image of a mixed snow-covered and snow-free scene from the spring 2016 flight. Image captured at 1/4000 s and ISO 100.*

[Figure]

*Figure 2: Image of a mixed snow-covered and snow-free scene from the spring 2016 flight. Image captured at 1/4000 s and ISO 100.*

[Figure]

*Figure 3: Image of a fully snow-covered scene from the winter 2016 flight. This flight occurred shortly after a snow fall event. Wind erosion and deposition features are well preserved in the surface of the winter snow pack, and well resolved by the camera. Image captured at 1/4000 s and ISO 400.*

*Finally, in line 3 of page 9, you highlight the calculation of error propagation as the most rigorous validation among them and a bonus of this study. In my opinion, is the actual snow probing that validates better and independently your dDSM, particularly because for your calculation you have only a sample of 6 CPs each as I understand, whereas you have 86 snow depth probing samples.*

**Response** Good point, this section has been re-worded to clarify. The key here is not necessarily that error propagation is the most rigorous approach, but that it should provide a good indication of uncertainty when no other data are available, an assumption that has been applied in previous recommendations in the literature (e.g., James et al., 2012). Ultimately, when deploying this technology for such application, relying on probing for each mission is not practical and arguably not desirable as it would defeat offered by the technique to some extent the advantage. In most case when repeating flights operationally, the uncertainty of the modelling can only be derived from an assessment of the AT. It should also be noted that the triangulations that provided the orthophotos and DSMs that were used for further analysis were fully constrained with all surveyed points used as control. The triangulation was then re-run with some points set as CPs to provide a conservative accuracy estimate, relative to the fully constrained solution. This has been clarified in section 3.2.2.

**3.3.1 Uncertainty associated with RPAS-derived snow depth**

*Please mention why did you assume that planimetric precision of each constituent DSM is negligible. In my experience, if you subtract two DSMs and there is just a slight misalignment, errors in the final dDSM (and bulk snow pack volume) can be considerable. That is why other studies have shown that co-georeferencing the DSMs using common GCPs can improve accuracy of the method.*

**Response** We agree that the planimetric contribution may not be (and often is not) negligible – that is a major focus of our discussion - we have re-worded this in an attempt to clarify. Our point here is that this assumption is inherent in only considering the contributions of vertical uncertainty of constituent DSMs to vertical uncertainty of dDSMs, which is an approach reported and recommended in the literature. Although we can only agree with the reviewer that common GCP across missions would be desirable, we must also stress that in our case it was not an option due to the inability to set permanent elevated markers on this protected conservation area. This and the lack of consistently exposed features such as rocks across missions dictated that a new GCP network needed to be setup, in turn justifying such careful assessment of repeatability and consistency. We have clarified this constraint in section 3.1.3.

**3.3.3 Repeatability of photogrammetric modelling**

*Please describe a little bit more what you have done in your classification algorithm.*

**Response** We have provided further detail in the text.

*What do you mean by dDSM$_3$? I can't find this DSM identifier anywhere else in the text/tables.*

**Response** Thank you, and we agree that this was ambiguous and have re-phrased to spring dDSM

**4. RESULTS**

**4.2.2 Assessment against reference probe data**

*It would be great if you can also display on map the difference between snow probed and RPAS estimated snow depth for each of the 86 sample points. That would give a spatial overview of the error, or across the slope for example. It could also open up for discussion of accuracy across slope and epochs or snow types.*

**Response** We agree that this could be an interesting analysis, and it was investigated initially as part of this research, but no clear spatial or terrain dependency was found (see Figure 4). We suspect that this is symptomatic of the shortcomings of the snow probe data, particularly the influence of vegetation, and potentially also geographic uncertainty in comparing high resolution datasets. The impact of slope and aspect on error with respect to field observations of snow depth deserves more attention, but would benefit most from a targeted field experiment.

[Figure]

*Figure 4: Scatter plots of residual between RPAS derived and probed snow depths and slope, elevation, and aspect.*

**4.2.3 Comparison of DSMs from independent RPAS flights**

*I would suggest merging Figure 6 and Figure 5 somehow. I think the manuscript has already too many Figures that can be reduced.*

**Response** This is a good suggestion and we have merged these figures into a single Figure 5.

5. *DISCUSSION*

*5.1 Performance of RPAS photogrammetry for resolving snow depth*

*I believe here you highlight well the main strength of your study; that is a relatively larger area and mapped at high resolution. This is interesting and perhaps your study should be more centered on this. However, you should highlight what are some of the limitations in your validations here.*

*Line 3, page 14: "Achievement of uncertainties <0.13 m": Please mention to which statistics you refer. I guess the propagation of error?.*

**Response** The text in both sections has been re-written to clarify, and the number adjusted to 0.14 m to reflect more accurate rounding. The value of 0.13 m comes from the empirical 90[th] percentile shown in Figure 6 and discussed previously in 4.2.3. As per our reply to an earlier comment, we reworked substantially the introduction and the merit of this study to highlight the fact that it applied to a relatively large area compared to previous studies.

*5.2.1 Vegetation*

*This has been highlighted by every study on snow depth mapping from RPAS, so should not be much of a novel finding discussion topic. I would however mention how it affected your dataset (as you already do), how it affected other studies and stress that solutions are really needed to solve this important caveat of the methodology.*

**Response** We have made some edits here to reflect these suggestions, particularly to stress the impact of potentially unreliable probe measurements on the validation, and the need for future work to characterise sub-snow vegetation processes in order to improve the applicability and reliability of this method.

*It would be interesting to know instead how did error varies with slope or with snow type spatially across your catchment area and across epochs (if any difference is noticeable).*

**Response** As noted previously and illustrated by additional figures in our response, there was no relationship observed between error relative to probed snow depths and terrain parameters. This is possibly a product of shortcomings in the reference dataset, as our other results provide an expectation of terrain dependent uncertainty. As such, no changes have been made to the text.

*5.2.2 Geo-location and co-registration*

*It would be good if the authors stress more clearly/directly/shortly what is the importance of all this and its relevance in a RPAS snow-mapping context. What I understand here is that the authors first want to point out that vertical dDSM uncertainties and errors increase when planimetric and horizontal geo-location errors are present, and particularly on steep slopes or presence of rock outcrops (not surprisingly). Then the aim to is to justify the found statistics in this study, particularly for the slope and this is interesting, but requires a more clearer explanation of the benefits and consequences.*

**Response** Understanding the impact of planimetric uncertainty on the quality of measurements obtained by DSM differencing is fundamental to characterising the uncertainty associated with surface change analysis. The importance of this increases with the advent of very high resolution datasets, such as those obtained from RPAS photogrammetry, and in areas of complex terrain. These effects have been identified previously (e.g., Nolan et al., 2015), but not characterised in the context of snow depth

mapping using RPAS photogrammetry. The characterisation we provide here should provide useful guidance on the potential limitations of the technique. We have amended the text to highlight the importance of this section.

*The main issue here is that is claimed that utilizing independent aero triangulation is better that cogeoreferencing the DSMs. I personally disagree with this statemet, co –georeferencing saves considerable time when snow free areas are available. And if not available, it's always easier to leave artificial features like high poles (seen all over the year), which require only 1 measurement for the entire seasonal snow cycle over multiple years. Many studies have reported that co-georeferencing their terrain and snow DSMs using common GCPs considerably improved the snow depth maps accuracy performance (because of all the problems you mentioned). They mitigate considerably all the issues of planimetric misalignment.*

**Response** We certainly agree that having permanent GCP markers installed in features to act as common GCPs throughout the year would provide desirable consistence with expected benefits on the accuracy of the dDSM. To us this process still leverages independent AT that should result in consistently co-registered DSMs "by design". However due to the status (Conservation Area) of our field site, this is simply not possible, a point that we did clarify in Section 3.1.3. Notwithstanding this, what we discuss here is the process of co-registering DSMs produced subsequently to AT, not processing AT that are inherently co-registered thanks to the use of common GCPs. This is an important difference and we invite the reviewer to reconsider his comment in view of this. To address this comment, we shall further describe and discuss below the four approaches that can be taken to producing well co-registered products. In view of this, we stand by our statements in Section 5.2.2:

1. The use of permanent, common GCP markers. A big advantage of this approach is in the time that can be saved in the field, although markers may still need to be cleared of snow and ice. Another advantage of this approach is that the contribution of GNSS uncertainty of GCP measurement to the total error budget will be constant for all triangulations. Drawbacks of this approach include the possibility that markers still need to be cleared of snow and/or ice prior to flights, negating time savings, and the potential for the introduction of a positive bias into surface elevations when markers are consistently elevated above ground level.

2. The use of GCPs placed in the field and surveyed for each measurement campaign, which is the approach we took. Where GCPs are surveyed in the appropriate coordinate system and related by a common benchmark (as in our study), the contribution to the total error budget of independently surveyed sets of GCPs will be minimised, and should be comparable to 1.

3. The use of "soft" GCPs, for which 3D coordinates are extracted for features from the triangulation of the first (reference) mission and then used as GCPs for subsequent missions. We evaluated this approach following our Autumn flight, but a combination of low contrast on exposed rock features, and snow obscuring features in later flights limited the utility of this approach. Furthermore, this approach will compound three sources of error for all subsequent DSMs, that of the initial GNSS GCP measurement, that of the reference triangulation, and that of the final triangulation.

4. Explicit co-registration of DSMs (e.g., Nolan et al., 2015) is what we discuss in the manuscript and represents an appealing solution because it can potentially minimise co-registration uncertainty substantially. While it may be appropriate for coarse models, where surface features are highly smoothed, we consider this method to be the least desirable, particularly for very high resolution products where spatial resolution and geo-location accuracy converge. The processes of applying a transformation between constituent DSM grids and resampling pixel values to the new grid will introduce a level of distortion and error to the co-registered product that may further degrade the snow depth signal that is being sought. Such degradation may in turn compromise the desired level of accuracy of the dDSM product.

*In addition, I would see this discussion to be more focused on snow itself, and its changes in aspect, and elevation in accordance with underlying terrain and how this will affect statistics of other people employing the method, under different snow landscapes. This rather than focusing on a general DSM context.*

**Response** We have modified this section to focus more directly on the relevance of measuring snow depth. The relationship between the magnitude of the residual between RPAS and probe derived snow depths and slope, aspect and elevation was assessed, but no clear dependency was revealed. This is possibly due to the quality of the probe measurements (e.g., impact of vegetation and sampling error) and in the case of elevation, the limited range of elevation sampled by probing (only along three elevation contours). We agree that the influence of terrain on uncertainty in derived snow depths is worthy of further investigation, but this would require a more targeted experiment.

*Finally, you mention that high quality GCPs are important, but you don't discuss the future of systems with on-board RTK which will probably substitute GCPs in the near future and can account more directly and precisely the errors you mention in roll, pitch and yaw by integrating IMU components.*

**Response** We agree that RTK equipped RPAS and "direct geo-referencing" are promising for improving the quality of output with reduced need of GCPs. With an absence of any GCPs, however, it becomes more difficult to confidently characterise triangulation quality, and the risk that gross errors may go un-detected is also increased. Additionally, RTK systems often suffer a weight and power consumption penalty that sees flight times reduced (e.g., an advertised 35 minute endurance of the Trimble UX5 HP, compared to the 50 minutes of the UX5 used here, https://www.trimble.com/agriculture/ux5). We have added these points to the manuscript.

*5.3 Pitfalls and limitation of RPAS photogrammetry*

*This large sub-chapter and more than three figures are dedicated to comparing the performance of two different black-box photogrammetric software's. I guess this is not part of your initial aims, and while is partially interesting for some communities, I think your goal is to outline a snow depth mapping method and not the performance of the particular software's. Therefore, I would personally reduce considerably this section. Particularly since inferences are made based on two different black-box software's and you don't really comment/outline/reference on the particular workflow of these products.*

**Response** We agree that there was scope to shorten this section, and have done so, including the removal of Figure 13, and merging of Figures 12 and 15. We however believe that this section has merit as it clearly demonstrates that commercially available and used software cannot always be relied upon for robust implementation and generation of high quality results, with the main metric used to assess AT being potentially misleading. The artefact detected is particularly problematic in the context of snow depth mapping, as when full snow cover exists over a study area, there is no straight-forward possibility to apply an empirical correction, and other approaches to remove such stripping (e.g., Fourier Transform) present a risk of loss of real signal. The mapping of snow depth is one of the few applications of RPAS photogrammetry where an entire surface may be transformed between measurements, making such issues particularly problematic to this application. We also believe that although this issue may affect only the one software we tested, the question it raises is significant and implications are far-reaching and relevant to others. We believe being able to document such issue and the process of its discovery, as well as our attempt to understand the source of the short-coming makes it relevant to the

increasing community of RPAS users to inform about such limitations. We have clarified this important point in the text.

*Another issue is that here you generalize as "limitations and pitfalls of RPAS photogrammetry" problems that were encountered within the particular software employed. Other studies have not reported these discussed issues with other software that I am aware (or they could not be verified?). Therefore, is difficult generalize such a discussion for snow depth mapping with RPAS.*

**Response** We agree that in this the artefact was a product of specific software and have made changes to the text to clarify. That said, it was fortuitous that the subtle error was discovered in the first place, and it is possible that it may have gone un-detected had each flight not had a near-identical flight path. It is possible that such errors appear in other datasets, yet they will become less obvious when flight paths differ between flights, or as surface complexity increases (e.g., due to either terrain, or vegetation). The take home message here is that these techniques demand vigilance, and should not be treated as "black box" solutions. As noted, we have amended the text to ensure that this key point is clear to readers.

*5.4 Spatial and temporal trends in snow cover*

*This section much better highlights the strengths of your study and more should be reflected in the introduction and aims. The same can be said about your conclusion. Nevertheless, more could be said about the high repeatability/change detection potential of your methodology, since your covered 2 distinct epochs.*

**Response** We thank the reviewer for this positive comment. We made our best to address the reviewer's constructive comments which helped us refine our focus and the strength of our study in the introduction in particular, and throughout the paper.

***Purely technical corrections at the very end ("technical corrections": typing errors, etc.).***

*Please make units consistent thorough the manuscript. Either cm or m (e.g. line 4 of page 6).*

**Response** This is a good suggestion and we have updated all cm units to m.

*In all tables and figures, please include full names and abbreviations. Often, only abbreviations are shown and the reader is forced to look for them in the text.*

**Response** Captions have been updated with definitions for abbreviations where necessary.

*Line 4 page 11: There is clearly a typographical error.*

**Response** This has been corrected.

*In figure 2 please add "points" to Ground Control...*

**Response** This has been corrected.

**2. Response to RC2 comments**
*This study presents results for a small watershed in New Zealand where repeat unmanned aircraft flights were used to map surface elevations using photogrammetric methods, and then snow depth via digital surface model differencing. There was one snow free flight, and two snow on flights, one winter and one spring. Although the snow depth results are presented, the main focus of the paper is more technically focused on methods, uncertainty, and validation.*

*The use of unmanned aerial systems in earth science is growing in popularity for good reason; the units are small, relatively inexpensive, easy to deploy, and the software to carry out structure from motion photogrammetry is becoming more accessible and user friendly. This study is a relevant and useful contribution to the growing body of literature using UAS to map snow depth and cryospheric processes at high resolution, fits within the scope of The Cryosphere, and should be accepted for publication after revisions. Following are broad recommendations that would improve the manuscript, namely in terms of readability and accessibility by a broader audience, particularly one that may not be familiar with mapping surface elevations/snow depth with UAS.*

**Response** We thank the reviewer for their positive comments. We have made a number of edits which we hope will improve the clarity and readability of the work, particularly for readers less familiar with the use of RPAS.

*- The acronym RPAS was new to me, likely a regional difference in terminology that I am unfamiliar with. In terms of search-ability I would suggest the switch to UAV or UAS (which is already used in the paper- so that would simplify things), or at minimum, mention the different terms use for unmanned aerial systems in the introduction and justify the use of RPAS rather than UAS.*

**Response** We prefer and have retained RPAS as this is the term used by relevant regulatory authorities, and is widely used within the surveying and geospatial community. RPAS also more accurately describes the operation of the UX5 (i.e., the flight is overseen by a human "pilot") and is a gender neutral term. We have, however, noted the use of synonymous terms unmanned aerial systems (UAS) and unmanned aerial vehicles (UAV) when the term RPAS is first introduced.

*- The manuscript reads as if the authors assume the reader has some understanding of photogrammetry, which is not necessarily a safe assumption. Something as simple as 'overlapping pictures are used to reconstruct a continuous 3 dimensional surface' very early on in the introduction would be helpful to provide context to the reader, and also making sure important terms are defined (like tie point). Also aerotriangulation is simply the georeferencing method by which ground control values are assigned to points, this could be defined once and then the term georeferencing could be used afterwards, which is a more accessible term. This paper dives into the technical very quick, but shouldn't forget to cover the basics, as well, since this is still a relatively new method for mapping snow depth.*

**Response** We agree that some of this material was potentially unclear to readers who are not familiar with photogrammetry and have amended the text to make it more accessible. In section 3.2.1., we now point out that the aero-triangulation is a means of georeferencing, but have retained the term aero-triangulation (AT) elsewhere in the text as it is an approach specific to photogrammetry.

*- This paper does a great job of covering uncertainty, but I think it is interesting and important to recognize the practical limitations of this method early on in the paper. It currently cannot scale up*

*beyond small watersheds for practical reasons, namely flight times and flight restrictions, which vary widely from country to country. Also setting out ground control points can be just as time consuming and limiting as carrying out snow surveys, which is why the authors themselves wanted to reduce the numbers of GCPs used per flight. Also vegetation is a critical issue in watersheds that have thick brush, or trees for that matter, so it is only useful and accurate in alpine watersheds. Discussing how these issues might be be overcome in the future to make this method operationally useful would be very interesting (i.e. that use of RTK on the UAS). As it stands snow in medium to large scale, and/or vegetated, watersheds can only be mapped with lidar, and while it is notably missing in the paper, repeat high resolution lidar flights for snow depth and SWE are being done in the Western US by the Airborne Snow Observatory at operationally relevant scales (https://doi.org/10.1016/j.rse.2016.06.018).*

**Response** We agree that the limitations are important and have added comments addressing these and have emphasised the demand for characterisation of vegetation compaction and the uncertainty that this introduces. Regarding onboard RTK, it is noted that the addition of RTK hardware to the payload typically comes at the expense of increased power demand and reduced flying time. Even with RTK, it is desirable to have some form of surveyed control marks in the study area in order to confidently characterise uncertainty and reduce the risk of gross errors being undetected. In the New Zealand context we benefit from the fact that almost all seasonal snow occurs above the treeline.

*-In the introduction the authors emphasize how valuable this method could be for understanding spatial variability in snow depth at high resolution, but then spend very little time actually presenting snow depth results for the two snow-on flights. I do think the uncertainty discussion is important and relevant, but so is the snow depth results, and more time should be spent on them. Also, snow water equivalent is only mentioned briefly at the end, this should be an entire results in the section and the measurement of densities should be covered in the methods. An estimate of SWE for the two flights would be really interesting.*

**Response** We agree that SWE comparisons are of great interest, and that is the focus of ongoing work, hence only briefly presented here. We have added more detail around measurement of densities, and the SWE results to this discussion point, but have kept this brief to maintain emphasis on assessment of the performance of RPAS photogrammetry, and the resolution of spatial variability in snow depth, which is the primary motivation for this work.

*(Minor note, on pg 9, line 29 the authors say the nominal accuracy for snow probes is +/- 1 cm, if this is from the literature it should have a citation, because I understand it to be much larger due mostly to user error, which they themselves recognize, in detail, later.)*

**Response** We agree that accuracy was not the best term in this case and have replaced it with precision – that being defined by the interval of graduations on the probes used. We agree that user error, and sampling uncertainty (e.g., presence of vegetation below the snow pack) may substantially limit the accuracy of probe measurements.

*-It is quite obvious that one of the authors has a thorough understanding of statistics. It gets tedious, and in these sections/figures most readers will just skip over. I would suggest for each relevant result adding 1 plain language summary before diving into the details to improve readability. 'Uncertainty is larger for more rapid changes in topography'.*

**Response** We have amended the text to improve clarity and bring forward the emphasis on key findings.

*-It is not clear to me why the authors spend so much space in terms of text and figures on georeferencing errors with older software when it could be covered in a few sentences, and more time could spent on more relevant results (i.e. the gist of this is that the old software had large errors, the new software performs better, so the old software should be avoided). This would also reduce the number of figures (there are so many).*

**Response** We believe the discussion of the relative performance of software is an important one, especially in the case of a relatively new methodology, such as RPAS photogrammetry, where commercially available software packages are commonly implemented. We have edited this section to make it more concise, and have reduced the number of figures here by two. We have merged the original Figures 12 and 15, and deleted Figure 13. As the reviewer rightly notes, such software packages are becoming increasingly accessible and widely used, but given the non-trivial nature of photogrammetry, we believe the emphasis on the significant impact of a small error is warranted. The error detected highlights a fundamental problem in the implementation of the photogrammetric solution in commercial software, emphasising the need for vigilance when using such products. The specific error detected here has the potential to be particularly problematic in applications of this methodology to snow studies, as there will often be no means to determine and apply an empirical correction.

*General editing comments: Writing structure and grammar need some attention, as they were notable enough to distract from the science being presented. The first paragraph of the Intro needs to be rewritten to read more consistently and should introduce the context and motivation for this study specifically. All paragraphs should be at least three sentences in length. There are many run on sentences that made reading and interpreting intent challenging. Watch out for the use of colloquial terms in a scientific context ('hamper' or 'impair' for something that is a challenge or difficult, the use of the word 'see' or 'saw' for things that don't have eyes). A small but related note, I associate the term epoch with geologic time scales (a division of time that is a subdivision of a period and is itself subdivided into ages, corresponding to a series in chronostratigraphy), I suggest not using this term and in most places through out the text it is unnecessary. For overall readability of the technical sections it might be useful to think about what content contributes to the overall purpose of the study given the audience (like equations 1-5, I don't find these critical to include, interested readers could be provided with a reference to follow up with). It maybe useful to have someone that is a physical scientist, but not involved in the study, read through the paper and give feedback.*

**Response** Thank you for the important feedback. We have made substantial edits to the text to improve clarity and readability. We have reduced the usage of epoch throughout the manuscript. We have retained equations 1-5 as they underscore some of the later findings and discussion points and are fundamental to the application of photogrammetry. This is addressed in more detail in response to reviewer one.

**References**

James, L. A., Hodgson, M. E., Ghoshal, S. and Latiolais, M. M. (2012): 'Geomorphic change detection using historic maps and DEM differencing: The temporal dimension of geospatial analysis', Geomorphology, 137(1), pp. 181–198. doi: 10.1016/j.geomorph.2010.10.039.

Nolan, M., Larsen, C. and Sturm, M. (2015): 'Mapping snow depth from manned aircraft on landscape scales at centimeter resolution using structure-from-motion photogrammetry', The Cryosphere, 9(4), pp. 1445–1463. doi: 10.5194/tc-9-1445-2015.

Sims, C. and Orwin, J. F. (2011): 'Snowmelt generation on a hydrologically sensitive mountain range, Pisa Range, Central Otago, New Zealand', Journal of Hydrology (NZ), 50(2), pp. 383–405.